# PLK4 drives centriole amplification and apical surface area expansion in multiciliated cells

Gina M LoMastro, Chelsea G Drown, Aubrey L Maryniak, Cayla E Jewett, Margaret A Strong, Andrew Jon Holland*

Department of Molecular Biology and Genetics, Johns Hopkins University School of Medicine, Baltimore, United States

**Abstract** Multiciliated cells (MCCs) are terminally differentiated epithelia that assemble multiple motile cilia used to promote fluid flow. To template these cilia, MCCs dramatically expand their centriole content during a process known as centriole amplification. In cycling cells, the master regulator of centriole assembly Polo-like kinase 4 (PLK4) is essential for centriole duplication; however recent work has questioned the role of PLK4 in centriole assembly in MCCs. To address this discrepancy, we created genetically engineered mouse models and demonstrated that both PLK4 protein and kinase activity are critical for centriole amplification in MCCs. Tracheal epithelial cells that fail centriole amplification accumulate large assemblies of centriole proteins and do not undergo apical surface area expansion. These results show that the initial stages of centriole assembly are conserved between cycling cells and MCCs and suggest that centriole amplification and surface area expansion are coordinated events.

## Editor's evaluation

PLK4 is the master regulator of centriole biogenesis, but whether it is also key for centriole amplification during differentiation of multiciliated cells (MCCs) has been questioned based on PLK4 chemical inhibition. In this fundamental study, using mouse models engineered to lack PLK4 or PLK4 activity, LoMastro et al., provide exceptional evidence that PLK4 and its activity are essential for centriole amplification in MCCs. Moreover, they show that centriole amplification in MCCs drives expansion of their apical surface. The findings settle the debate whether PLK4 is crucial for centriole amplification in multiciliated cells and will determine how cell biologists and experts interested in multi-ciliogenesis-related pathologies understand this process.

*For correspondence: aholland@jhmi.edu

## Introduction

Multiciliated cells (MCCs) are specialized epithelial cells that line the surface of the respiratory tract, brain ventricles, and reproductive systems. MCCs contain tens to hundreds of motile cilia that collectively beat to drive fluid flow across epithelial surfaces. Each motile cilium is templated by a centriole or basal body that docks at the plasma membrane and extends a ciliary axoneme. Abnormalities in motile cilia assembly or function cause several pathologies, including chronic respiratory tract infections, hydrocephalus, and reduced fertility (*Reiter and Leroux, 2017*; *Spassky and Meunier, 2017*; *Terré et al., 2019*; *Yuan et al., 2019*).

In cycling cells, centriole duplication is tightly controlled so that a single new procentriole is created on the wall of each of the two parent centrioles during every cell cycle (*Nigg and Holland, 2018*). By contrast, MCC progenitors with two parent centrioles produce tens to hundreds of new centrioles

**eLife digest** Every day, we inhale thousands of viruses, bacteria and pollution particles. To protect against these threats, cells in our airways produce mucus that traps inhaled particles before they reach the lungs. This mucus then needs to be removed to prevent it from becoming a breeding ground for microbes that may cause a respiratory infection. This is the responsibility of cells covered in tiny hair-like structures called cilia that move together to propel the mucus-trapped particles out of the airways.

These specialized cells can have up to 300 motile cilia on their surface, which grow from structures called centrioles that then anchor the cilia in place. Multiciliated cells are generated from precursor cells that only have two centrioles. Therefore, as these precursors develop, they must produce large numbers of centrioles, considerably more than other cells that only need a couple of extra centrioles during cell division. However, recent studies have questioned whether the precursors of multiciliated cells rely on the same regulatory proteins to produce centrioles as dividing cells.

To help answer this question, LoMastro et al. created genetically engineered mice that lacked or had an inactive form of PLK4, a protein which controls centriole formation in all cell types lacking multiple cilia. This showed that multiciliated cells also need this protein to produce centrioles. LoMastro et al. also found that multiciliated cells became larger while building centrioles, suggesting that this amplification process helps control the cell's final size.

Defects in motile cilia activity can lead to fluid build-up in the brain, respiratory infections and infertility. Unfortunately, these disorders are difficult to diagnose currently and there is no cure. The findings of LoMastro et al. further our understanding of how motile cilia are built and maintained, and may help future scientists to develop better diagnostic tools and treatments for patients.

during their differentiation to generate the basal bodies required for multiciliogenesis. MCCs utilize two pathways to promote massive centriole amplification: a 'centriole pathway' where multiple procentrioles assemble on the wall of the two parent centrioles, and a 'deuterosomal pathway' where dozens of specialized structures called deuterosomes provide the nucleation sites for multiple procentrioles (*Sorokin, 1968*; *Steinman, 1968*; *Brenner, 1969*; *Kalnins and Porter, 1969*; *Anderson and Brenner, 1971*; *Klos Dehring et al., 2013*; *Al Jord et al., 2014*; *Revinski et al., 2018*). MCC differentiation is defined by distinct stages (*Al Jord et al., 2017*). In mouse ependymal cells, procentrioles form around parent centrioles and deuterosomes during amplification (A) phase. Procentrioles elongate during growth (G) phase and detach from their growing platforms at disengagement (D) phase. Disengaged centrioles migrate and dock at the apical membrane to become basal bodies that form motile cilia during the multiple basal body (MBB) phase.

How MCCs control the final number of centrioles generated remains an open question. Centriole number scales with apical surface area in multiciliated mTECs and *Xenopus* embryonic epidermis (*Nanjundappa et al., 2019*; *Kulkarni et al., 2021*). In *Xenopus*, mechanical stretching forces help couple centriole number with MCC apical area (*Kulkarni et al., 2021*). However, while mouse MCCs amplify centrioles in a single round of centriole biogenesis, *Xenopus* epidermal MCCs produce centrioles in two distinct waves (*Al Jord et al., 2014*; *Kulkarni et al., 2021*). Thus, it remains to be established whether similar or distinct mechanisms operate to set centriole numbers in mammalian MCCs.

Polo-like kinase 4 (PLK4) is the master regulator of centriole biogenesis and is required for centriole assembly in all cycling mammalian cell types studied (*Bettencourt-Dias et al., 2005*; *Habedanck et al., 2005*). During centriole duplication, CEP152 and CEP192 recruit PLK4 to the surface of the parent centrioles (*Cizmecioglu et al., 2010*; *Hatch et al., 2010*; *Kim et al., 2013*; *Sonnen et al., 2013*; *Park et al., 2014*). PLK4 kinase activity directs the assembly of STIL and SAS6 into a central cartwheel that provides the structural foundation for the new procentriole, and inhibition of PLK4 activity blocks centriole assembly (*Dzhindzhev et al., 2014*; *Ohta et al., 2014*; *Kratz et al., 2015*; *Moyer et al., 2015*). PLK4 is recruited to the surface of both parent centrioles and deuterosomes in MCCs (*Nanjundappa et al., 2019*). However, recent work has shown that inhibiting PLK4 kinase activity with the ATP-competitive small molecule centrinone has no impact on centriole amplification in mouse multiciliated ependymal cells or mTECs (*Nanjundappa et al., 2019*; *Zhao et al., 2019*). In addition, shRNA-mediated depletion of PLK4 modestly delays centriole assembly in mTECs but

does not reduce the final number of centrioles produced or the fraction of cells forming motile cilia (***Nanjundappa et al., 2019***). These data suggest there are distinct requirements for PLK4 in promoting centriole assembly in cycling cells and differentiating mouse MCCs. However, PLK4 levels are upregulated ~20-fold in differentiating mTECs, and it remains plausible that partial inhibition or depletion of PLK4 is not sufficient to prevent centriole amplification in MCCs (***Hoh et al., 2012***).

In this study, we use genetic approaches to define the requirements of PLK4 protein and kinase activity for centriole amplification in MCCs. These genetic tools have the advantage of achieving highly specific and penetrant target engagement. In addition, we investigate the impact of blocking centriole amplification on the differentiation and surface area scaling of MCCs. Our data show that PLK4 protein and kinase activity are required for centriole assembly in mouse MCCs, demonstrating that the early stages of centriole assembly are similar in cycling cells and MCCs. Moreover, we show that centriole amplification drives an increase in apical surface area in airway MCCs, suggesting that increased centriole number is a central driver of surface area scaling in MCCs.

## Results
### Generation of a conditional loss-of-function allele of PLK4

To test the requirement of PLK4 for centriole amplification in MCCs, we created mice with exon 5 of the *Plk4* gene flanked by *LoxP* recombination sites (hereafter *Plk4^F* allele). Cre-mediated excision of exon 5 generates a *Plk4* allele (hereafter *Plk4^Δ* allele) lacking a large portion of the PLK4 kinase domain, including the DFG motif required for kinase activity. Splicing of *Plk4* exon 4 into exon 6 is predicted to generate a frameshift that creates a premature termination codon at amino acid 119 (***Figure 1—figure supplement 1A***).

Intercrosses of *Plk4^F/Δ* mice produced *Plk4^F/F* and *Plk4^F/Δ* progeny but *Plk4^Δ/Δ* animals were never observed (***Figure 1—figure supplement 1B***). To validate the *Plk4^F* allele, we inactivated PLK4 throughout the central and peripheral nervous system using Nestin^Cre. *Plk4^F/F*;Nestin^Cre animals developed severe microcephaly leading to a 35% reduction in telencephalic area at E14.5 (***Figure 1—figure supplement 1C, D***). To examine how the loss of PLK4 disrupted centriole duplication in vivo, we analyzed dividing neural progenitor cells (NPCs) at the ventricular zone in E14.5 cortices. As expected, loss of PLK4 led to a dramatic decrease in centriole number (***Figure 1—figure supplement 1E***). We conclude that *Plk4^F* is a conditional loss-of-function allele of *Plk4*.

### Ependymal cells lacking PLK4 fail to amplify centrioles

To test the impact of PLK4 loss in MCCs, *Plk4^F/F* and *Plk4^F/+* ependymal cells were transduced with adeno-associated-virus (AAV) expressing NLS-Cre-GFP (hereafter AAV-Cre). Ependymal cells were induced to differentiate and transduced cells expressing nuclear GFP were analyzed at various time points after infection (***Figure 1A***, ***Figure 1—figure supplement 1F***). The frequency of FOXJ1+ ependymal cells at each time point was similar in *Plk4^F/+* and *Plk4^F/F* Cre+ cells (***Figure 1B***). In addition, *Plk4^F/F* and *Plk4^F/+* cells formed deuterosomes at the same frequency (***Figure 1C***), indicating that loss of PLK4 does not block the differentiation of MCCs in vitro.

The PLK4 receptor CEP152 stained the surface of parent centrioles and deuterosomes in ependymal cells with or without PLK4 (***Figure 2—figure supplement 1A***). To determine the extent of PLK4 depletion, we quantified the intensity of PLK4 on the deuterosomes of differentiating MCCs. PLK4 abundance was reduced by 80% in Cre+ *Plk4^F/F* ependymal cells compared to Cre+ *Plk4^F/+* control cells (***Figure 1D and E***). Consistent with PLK4 loss, deuterosomal recruitment of the PLK4 substrate and early centriole assembly factor STIL was abolished in Cre-expressing *Plk4^F/F* ependymal cells (***Figure 1F and G***). These data show that similar to cycling cells, PLK4 is required to recruit STIL to the site of procentriole assembly in differentiating ependymal cells.

To evaluate the effect of PLK4 loss on centriole amplification in ependymal cells, we analyzed centriole number in disengagement phase (D phase) when individual centrioles can be resolved. Cre+ *Plk4^F/+* ependymal cells amplified ~100 centrioles per cell, while most *Plk4^F/F* ependymal cells failed to undergo centriole amplification and contained only the two parent centrioles that were present at the start of MCC differentiation (***Figure 2A and B***). Consistently, Cre+ FOXJ1+*Plk4^F/F* ependymal cells rarely nucleated multicilia (***Figure 2C and D***, ***Figure 2—figure supplement 1B, C***).

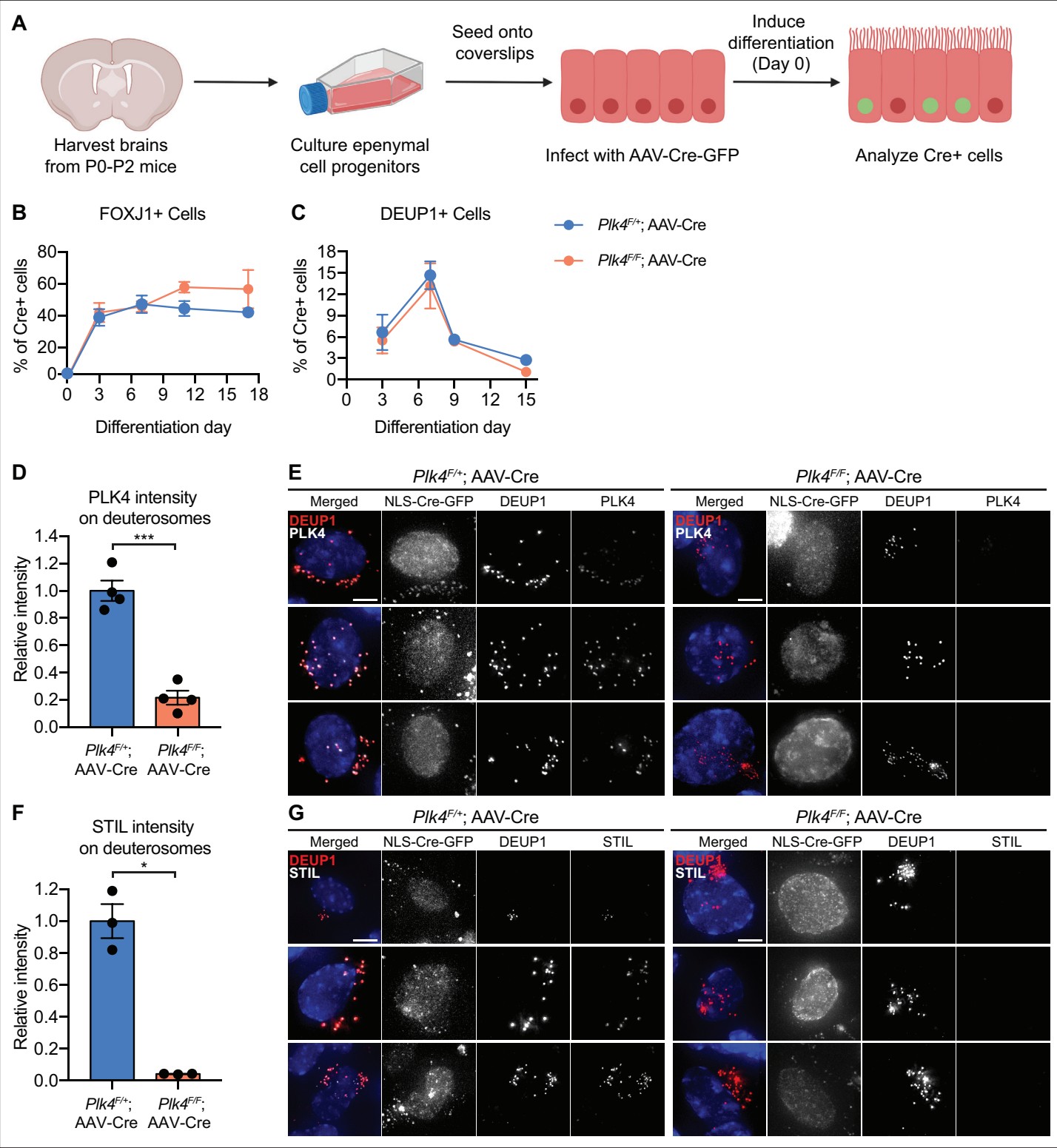

**Figure 1.** Ependymal cells lacking PLK4 differentiate but fail to recruit STIL to deuterosomes. (**A**) Schematic overview of the experiment to recombine the *Plk4^F/F* allele in ependymal cells in vitro. Ependymal cells were harvested from *Plk4^F/F* mice or *Plk4^F/+* controls and seeded onto coverslips. On differentiation day 0, the ependymal cells were transduced with AAV-Cre in serum-free media. (**B**) Quantification of the percent of FOXJ1-expressing ependymal cells transduced with AAV-Cre-GFP. N=3–4 brains, n>300 cells. Two-tailed Welch's t-test. (**C**) Quantification of the percent of AAV-Cre-GFP transduced ependymal cells with deuterosomes, using DEUP1 as a maker. Day 0 N=3, day 7 N=5, day 9 N=2, day 15 N=3; n>300 cells. Two-tailed Welch's t-test. (**D**) Quantification of the intensity of PLK4 on deuterosomes in ependymal cells on differentiation day 7. Cells in all stages of centriole

*Figure 1 continued on next page*

*Figure 1 continued*

amplification were measured and pooled. N=4, n≥20. Two-tailed Welch's t-test. (**E**) Confocal images of *Plk4^F/F^* or *Plk4^F/+^* ependymal cells transduced with AAV-Cre-GFP. Cells were stained with DEUP1 to mark deuterosomes (red) and PLK4 (white). Scale bar=5 µm. (**F**) Quantification of the intensity of STIL on deuterosomes in ependymal cells on differentiation day 7. Cells in all stages of centriole amplification were measured and pooled. N=3, n≥30. Two-tailed Welch's t-test. (**G**) Confocal images of *Plk4^F/F^* or *Plk4^F/+^* ependymal cells transduced with AAV-Cre-GFP. Cells were stained with DEUP1 (red) and STIL (white). Scale bar=5 µm. Data information: All data represent the means ± SEM. *p<0.05; ***<0.001.

The online version of this article includes the following source data and figure supplement(s) for figure 1:

**Source data 1.** Values for biological and technical replicates for graphs in *Figure 1*.

**Figure supplement 1.** Knockout of PLK4 leads to centriole loss in neural progenitor cells (NPCs).

**Figure supplement 1—source data 1.** Values for biological and technical replicates for graphs in *Figure 1—figure supplement 1*, and pedigree information for *Figure 1—figure supplement 1B*.

To confirm that centriole assembly was blocked in ependymal cells lacking PLK4, we performed expansion microscopy on AAV-Cre transduced *Plk4^F/+^* and *Plk4^F/F^* ependymal cells. We observed procentrioles associated with both deuterosomes and parent centrioles in control cells, but procentrioles were absent in ependymal cells lacking PLK4 (*Figure 2E and F*). This confirms that PLK4 is required for the earliest stages of centriole assembly in ependymal MCCs.

Multiciliated ependymal cells are derived from a subset of NPCs. Loss of centrioles in NPCs activates a USP28-53BP1-TP53 mitotic surveillance pathway that blocks NPC proliferation and indirectly depletes MCCs (*Lambrus et al., 2016*; *Phan et al., 2021*). To enable the proliferation of NPCs lacking PLK4 and centrioles, we inactivated the mitotic surveillance pathway in *Plk4^F/F^*;Nestin^Cre^ mice through knockout of USP28. While *Plk4^F/F^*;Nestin^Cre^ mice died shortly after birth, *Plk4^F/F^*;*Usp28^F/F^*;Nestin^Cre^ animals survived past postnatal day 21. FOXJ1+ ependymal cells could readily be identified in the lateral ventricles of *Plk4^F/F^*;*Usp28^F/F^*;Nestin^Cre^ brains. However, these cells lacked CEP164+ basal bodies and were devoid of multicilia (*Figure 3*). Taken together, these data show that PLK4 is required for centriole amplification in multiciliated ependymal cells in vitro and in vivo.

## PLK4 is required for centriole amplification in mouse trachea epithelial cells

In mouse trachea epithelial cells (mTECs) centriole amplification is classified into six stages (*Zhao et al., 2013*). mTECs begin with two parent centrioles (Stage I) that amplify centrioles together with deuterosomes (Stage II). Deuterosomes enlarge and are encircled by multiple procentrioles in a flower-like arrangement (Stage III). Procentrioles elongate (Stage IV) and are detached from deuterosomes and parent centrioles (Stage V). Finally, centrioles dock at the apical membrane and initiate the assembly of muliticilia (Stage VI). To test the requirement of PLK4 for centriole amplification in mTECs, we isolated mTECs from *Plk4^F/F^*;Rosa26^CreERT2^ animals or littermate *Plk4^F/+^*;Rosa26^CreERT2^ controls (hereafter *Plk4^F/F^*;R26^Cre^ or *Plk4^F/+^*;R26^Cre^). Some animals also contained a Centrin-GFP transgene to mark centrioles (indicated in Figures as Centrin-GFP). Cre recombination was induced in vitro with 4OHT 2 days before exposure to an air-liquid interface (ALI) and cells were analyzed at various time points thereafter (*Figure 4A*).

Comparable fractions of *Plk4^F/F^*;R26^Cre^ and *Plk4^F/+^*;R26^Cre^ mTECs expressed FOXJ1 at ALI 7 and ALI 21 and formed deuterosomes at ALI 3 and ALI 21 (*Figure 4B and C*). Similar to ependymal cells, CEP152 was recruited to deuterosomes in both *Plk4^F/F^*;R26^Cre^ and control mTECs (*Figure 4—figure supplement 1A*). The percentage of cells with deuterosomes decreased from ~25% at ALI 3 to ~5% at ALI 21 in both control and *Plk4^F/F^*;R26^Cre^ mTECs (*Figure 4C*). These data show that mTECs lacking PLK4 undergo similar transitions to cells that amplify centrioles: first recruiting CEP152 to deuterosomes and later degrading/disassembling deuterosomes.

In control mTECs, PLK4 localized to deuterosomes during the early stages of centriole amplification. Deuterosomal levels of PLK4 declined after centriole disengagement, and weak PLK4 staining was retained on disengaged centrioles (*Figure 4—figure supplement 1B*). About 55% of control mTECs exhibited PLK4 staining on deuterosomes at ALI 3, reflecting the percentage of DEUP1+ cells in the early stages of amplification. However, PLK4 was only present on deuterosomes in 15% of *Plk4^F/F^*;R26^Cre^ mTECs at ALI 3 (*Figure 4D and E*). Consistent with the loss of PLK4, the deuterosomal recruitment of the cartwheel proteins STIL and SAS6 was impaired in *Plk4^F/F^*;R26^Cre^ mTECs (*Figure 4F*

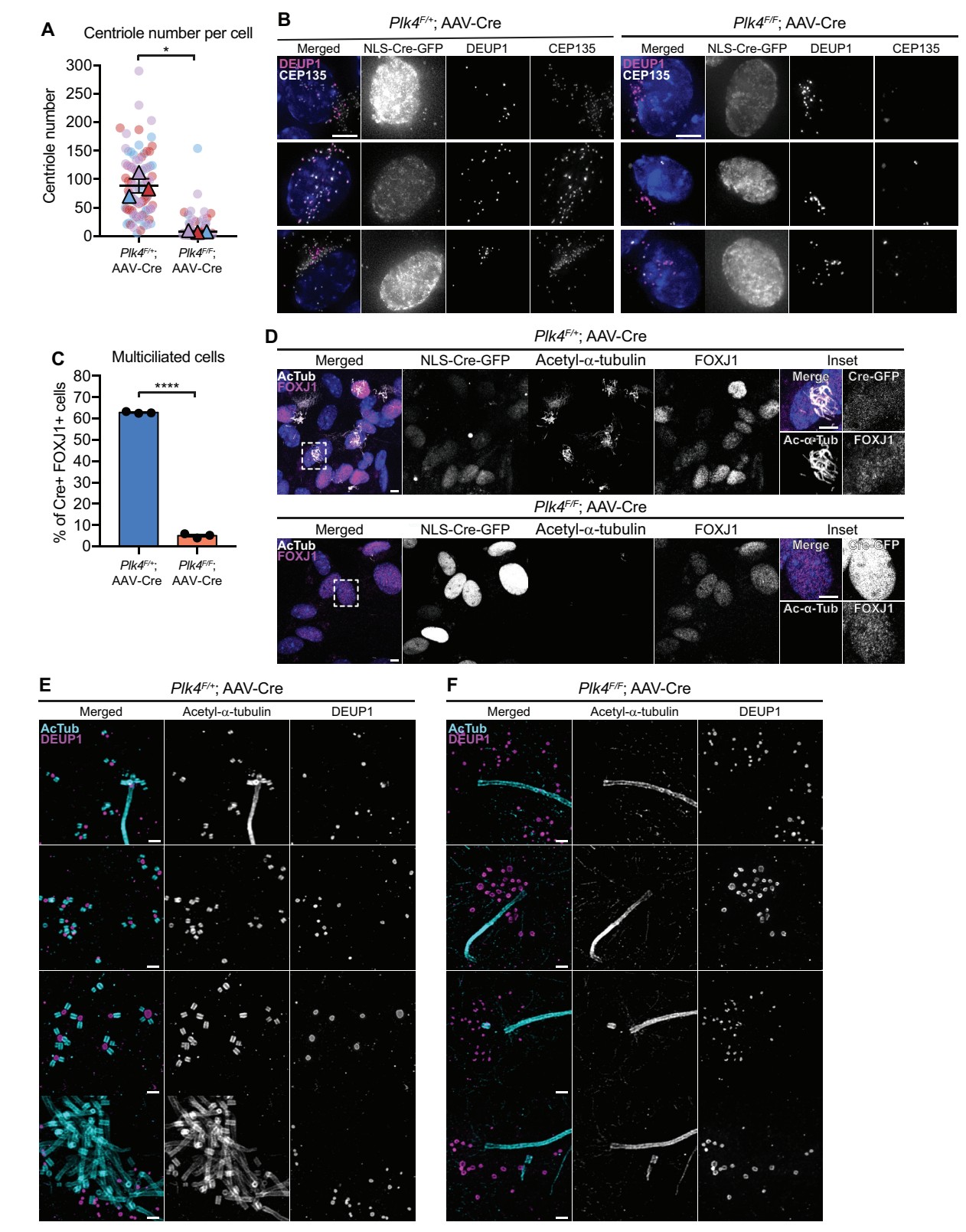

**Figure 2.** Loss of PLK4 blocks centriole assembly in ependymal cells. (**A**) Quantification of the number of centrioles per cell in disengagement (D) phase. *Plk4^{F/F}* or *Plk4^{F/+}* ependymal cells transduced with AAV-Cre-GFP were analyzed at differentiation days 7 and 8. N=3, n≥20. Two-tailed Welch's t-test. (**B**) Images of *Plk4^{F/F}* or *Plk4^{F/+}* ependymal cells transduced with AAV-Cre-GFP. Cells were stained with DEUP1 (magenta) and CEP135 to mark centrioles (white). Scale bar=5 µm. (**C**) Quantification of the percent of FOXJ1+*Plk4^{F/F}* or *Plk4^{F/+}* ependymal cells transduced with AAV-Cre-GFP that have multiple

*Figure 2 continued on next page*

*Figure 2 continued*

cilia at differentiation day 12. Acetylated tubulin staining was used to identify multiciliated cells. N=3, n>300. Two-tailed Welch's t-test. (**D**) Confocal images of *Plk4^F/F^* or *Plk4^F/+^* ependymal cells transduced with AAV-Cre-GFP. Cells were stained with antibodies against FOXJ1 (magenta) and acetylated-α-tubulin (white). Scale bar=5 µm. (**E, F**) Expansion microscopy of *Plk4^F/F^* or *Plk4^F/+^* ependymal cells at differentiation day 5 transduced with AAV-Cre-GFP. Cells were stained with acetylated-α-tubulin (cyan) and DEUP1 (magenta). Scale bar=0.5 µM. Data information: All data represent the means ± SEM. **p<0.01; ****<0.0001.

The online version of this article includes the following source data and figure supplement(s) for figure 2:

**Source data 1.** Values for biological and technical replicates for graphs in *Figure 2*.

**Figure supplement 1.** Ependymal cells lacking PLK4 recruit CEP152 but fail centriole amplification.

**Figure supplement 1—source data 1.** Values for biological and technical replicates for graphs in *Figure 2—figure supplement 1*.

*and G*; *Figure 4—figure supplement 1C, D*). Moreover, while 40% of control mTECs formed multiple cilia, only 4% of *PLK4^F/F^*;R26^Cre^ mTECs contained CEP164+ basal bodies and nucleated multiple cilia by ALI 7 (*Figure 4H, I*).

To ensure that centriole amplification and ciliogenesis were blocked and not simply delayed by the loss of PLK4, we analyzed mTECs at ALI 21. Although ~25% of both control and *PLK4^F/F^*;R26^Cre^ mTECs expressed FOXJ1 (*Figure 4B*), fewer than 5% of *PLK4^F/F^*;R26^Cre^ mTECs underwent centriole amplification at ALI 21 (*Figure 4J and K*). The small fraction of *Plk4^F/F^*;R26^Cre^ cells that underwent centriole amplification are likely to have escaped Cre-mediated recombination and consequently express PLK4 at normal levels. We conclude that PLK4 is also required for centriole amplification in mTECs.

## mTECs that fail centriole amplification accumulate assemblies of centriole components

Control mTECs frequently formed irregularly shaped assemblies of the distal centriole protein Centrin at the early stages of centriole amplification. However, these structures were mostly absent in fully differentiated MCCs (*Figure 5—figure supplement 1A*). By contrast, 4OHT-treated *Plk4^F/F^*;R26^Cre^ cells that failed centriole amplification formed large filamentous assemblies of Centrin (hereafter PLK4 knockout) (*Figure 5A and B*). To analyze these assemblies in greater detail, we performed expansion microscopy on control and PLK4 knockout mTECs at ALI 3. In control mTECs, amplified procentrioles were stained with acetylated-α-tubulin to mark the centriole microtubule wall, and Centrin which resides in the lumen of the centriole (*Figure 5C*). In PLK4 knockout mTECs, acetylated-α-tubulin and

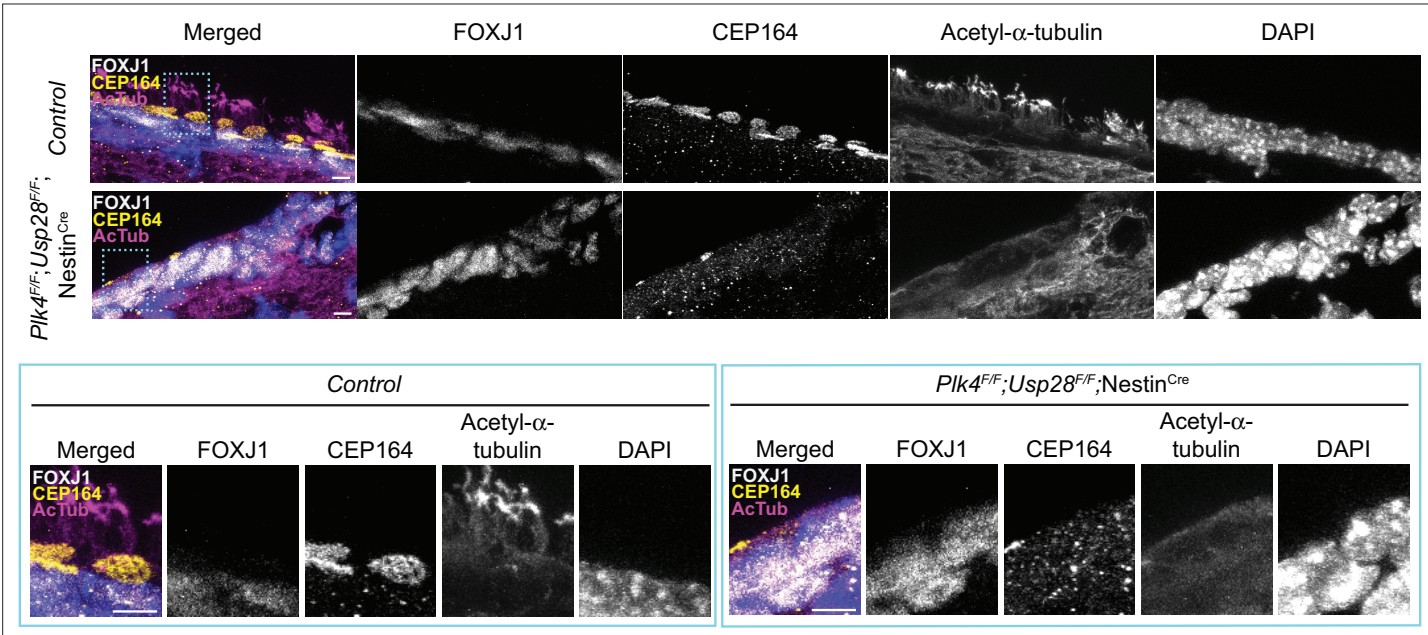

**Figure 3.** PLK4 is required for multicliogenesis in vivo. Confocal images of the lateral ventricle from control or *Plk4^F/F^*;*Usp28^F/F^*;Nestin^Cre^ mouse brains at P21. Sections were stained with antibodies against FOXJ1 (white), CEP164 (yellow), and acetylated-α-tubulin (magenta). Scale bar=5 µm.

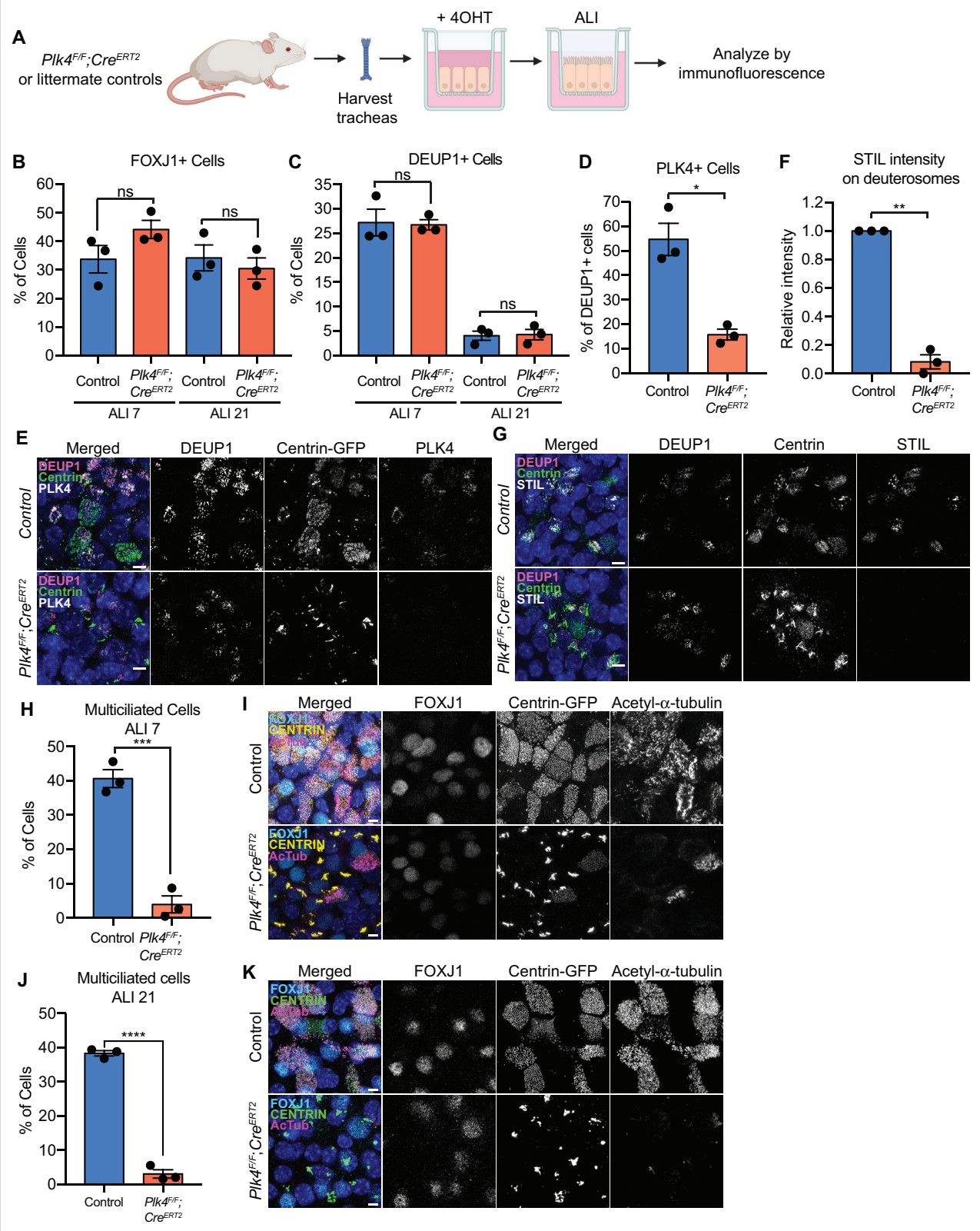

**Figure 4.** PLK4 promotes centriole assembly in mTECs. (**A**) Schematic of the experimental design to genetically deplete PLK4 in mTECs. *Plk4^F/F^;R26^Cre^* tracheal epithelial cells were seeded onto collagen-coated transwell filters and allowed to proliferate until confluence (day 3). At this point, Cre-mediated recombination of the *Plk4^F/F^* allele was induced by the addition of 4OHT to the media. After 2 days, mTECs were exposed to ALI to initiate differentiation. (**B**) Quantification of the percent of control or *Plk4^F/F^;R26^Cre^* mTECs expressing FOXJ1 at ALI 7 and ALI 21. N=3, n>500. Two-tailed Welch's

*Figure 4 continued on next page*

*Figure 4 continued*

t-test. (**C**) Quantification of the percent of control or *Plk4^F/F^*;R26^Cre^ mTECs with deuterosomes at ALI 7 and ALI 21. N=3, n>500. Two-tailed Welch's t-test. (**D**) Quantification of the percent of control or *Plk4^F/F^*;R26^Cre^ mTECs that express PLK4 at ALI 3. N=3, n>500. Two-tailed Welch's t-test. (**E**) Confocal images of control and *Plk4^F/F^*;R26^Cre^ mTECs at ALI 3. Cells were expressing Centrin-GFP and stained with DEUP1 (magenta) and PLK4 (white). Scale bar=5 μm. (**F**) Quantification of the intensity of STIL on deuterosomes in control or *Plk4^F/F^*;R26^Cre^ mTECs at ALI 3. Cells in all stages of centriole amplification were measured and pooled. N=3, n>500. Two-tailed Welch's t-test. (**G**) Confocal images of control and *Plk4^F/F^*;R26^Cre^ mTECs at ALI 3. Cells were expressing Centrin-GFP and stained with DEUP1 (magenta) and STIL (white). Scale bar=5 μm. (**H**) Quantification of multiciliated mTECs at ALI 7. N=3, n>500. Two-tailed Welch's t-test. (**I**) Confocal images of control and *Plk4^F/F^*;R26^Cre^ mTECs at ALI 7. Cells were expressing Centrin-GFP (yellow) and stained with FOXJ1 (cyan) and acetylated-α-tubulin (magenta). Scale bar=5 μm. (**J**) Quantification of multiciliated mTECs at ALI 21. N=3, n>500. Two-tailed Welch's t-test. (**K**) Confocal images of control and *Plk4^F/F^*;R26^Cre^ mTECs at ALI 21. Cells were expressing Centrin-GFP (green) and stained with FOXJ1 (cyan) and acetylated-α-tubulin (magenta). Scale bar=5 μm. Data information: All data represent the means ± SEM. *p<0.05; **<0.01; ***<0.001; ****<0.0001 and not significant indicates p>0.05. ALI, air-liquid interface; mTEC, mouse trachea epithelial cell.

The online version of this article includes the following source data and figure supplement(s) for figure 4:

**Source data 1.** Values for biological and technical replicates for graphs in *Figure 4*.

**Figure supplement 1.** mTECs lacking PLK4 recruit CEP152 but fail centriole amplification.

**Figure supplement 1—source data 1.** Values for biological and technical replicates for graphs in *Figure 4—figure supplement 1*.

Centrin marked the parent centrioles (*Figure 5D*, arrows) and the filamentous Centrin assemblies. Consistent with the observations in ependymal cells, procentrioles were not observed in mTECs lacking PLK4.

To investigate the physical properties of the Centrin assemblies, we performed fluorescence recovery after photobleaching (FRAP) in PLK4 knockout and control mTECs expressing Centrin-GFP. While Centrin-GFP recovered by 23% over 10 min at the amplified centrioles of control mTECs, recovery was limited to 10% in the filamentous assemblies formed in mTECs lacking PLK4 (*Figure 5—figure supplement 1B, C*).

We examined the localization of several other centriole and pericentriolar material (PCM) proteins in mTECs lacking PLK4. The proximal centriole protein CEP135 formed large spherical assemblies in PLK4 knockout mTECs. These assemblies were distinct from the filamentous assemblies of Centrin observed in mTECs that fail centriole amplification (*Figure 5B*). The distal centriole protein Centrobin, the distal appendage component CEP164 and γTubulin also formed ectopic assemblies in mTECs lacking PLK4 (*Figure 5E–G*). However, the centriole cartwheel components STIL and SAS6, and the PCM component CDK5RAP2 did not form detectable assemblies in *PLK4^F/F^*;R26^Cre^ mTECs (*Figure 5G*, *Figure 4—figure supplement 1C*, *Figure 5—figure supplement 1D*).

To determine whether the assemblies of centriole proteins are caused by a failure of centriole amplification, we examined the effect of knocking out the centriole component SAS4 in mTECs. At ALI 7–35% of control *Sas4^F/+^*;R26^Cre^ mTECs were multiciliated, while ~18% of *Sas4^F/F^*;R26^Cre^ mTECs formed MCCs (*Figure 5—figure supplement 1E, F*). At the same time point, 15% of *Sas4^F/F^*;R26^Cre^ mTECs displayed filamentous assemblies of Centrin and spherical assemblies of CEP135, a penetrance similar to that observed in mTECs lacking PLK4 (*Figure 5—figure supplement 1G*). These data suggest that the elevated expression of centriole proteins in mTECs drives the formation of assemblies of specific centriole components when centriole amplification is prevented.

## Evaluating centrinone as a tool to inhibit PLK4 kinase activity in mouse MCCs

We next investigated the requirement of PLK4 kinase activity for centriole amplification in MCCs. A previous study reported that treatment with 1.5 μM of the PLK4 inhibitor centrinone failed to block procentriole production in ependymal cells (*Zhao et al., 2019*). Given that this concentration of centrinone may be insufficient to fully inhibit PLK4 in MCCs, we treated differentiating ependymal cells with a broad range of centrinone concentrations (1–10 μM). After 5 days, PLK4 abundance on deuterosomes was measured as a readout of PLK4 inhibition (*Holland et al., 2010*). The levels of PLK4 progressively increased in ependymal cells treated with ≥2.5 μM centrinone (*Figure 6—figure supplement 1A*). At doses above 2.5 μM, we also observed a progressive decline in the fraction of MBB and disengagement (D) phase cells and an increase in the fraction of cells in amplification (A) or growth (G) phase (*Figure 6—figure supplement 1B, C*). Moreover, cells that reached MBB phase in the presence of 5 μM or 10 μM centrinone contained fewer centrioles than controls (*Figure 6—figure*

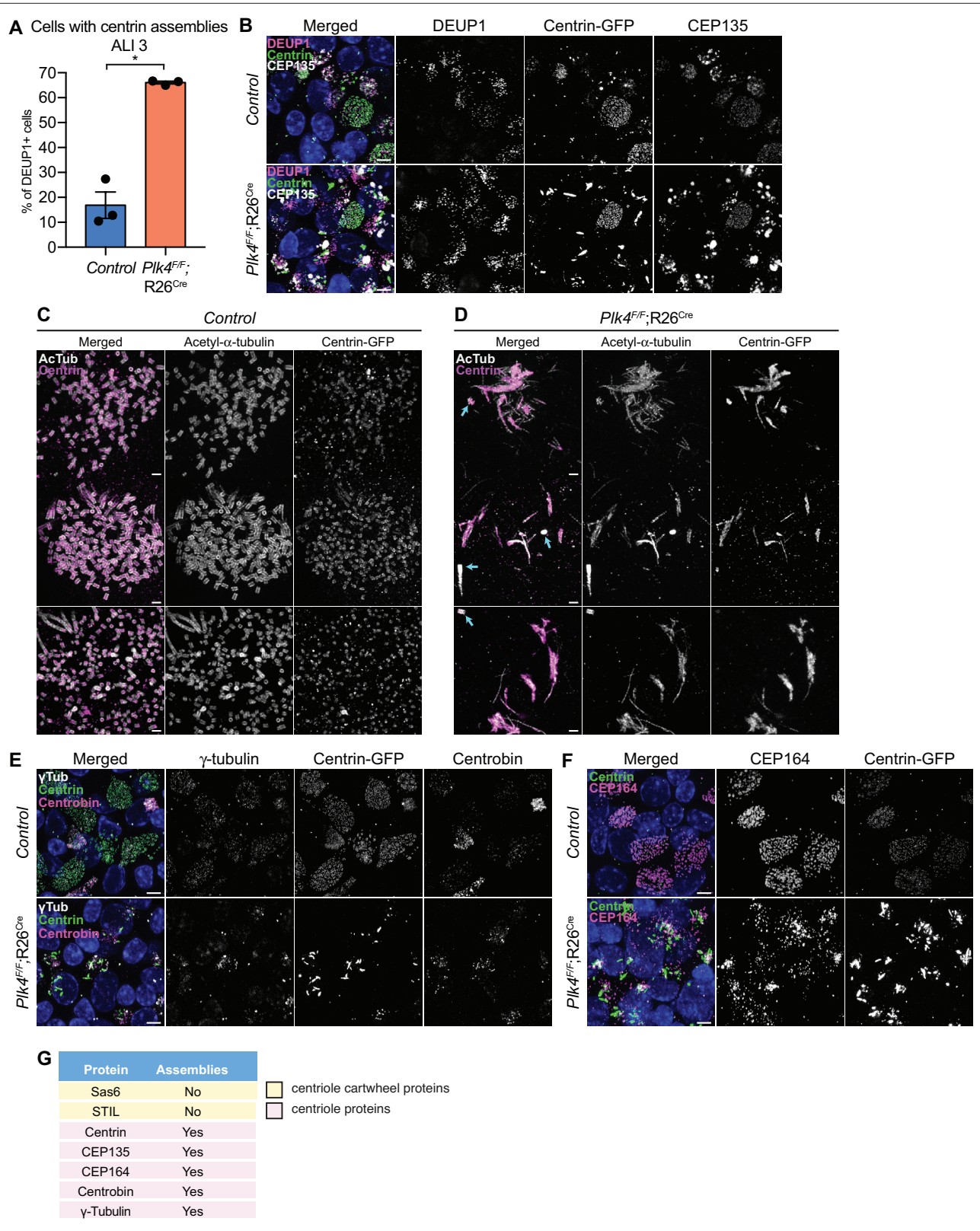

**Figure 5.** mTECs that fail centriole amplification form aberrant assemblies of centriole proteins. (**A**) Quantification of the percent of control and *Plk4*F/F;R26Cre mTECs with Centrin aggregates at ALI 3. N=3, n>150. Two-tailed Welch's t-test. (**B**) Confocal images of control and *Plk4*F/F;R26Cre mTECs at ALI 3. Cells were expressing Centrin-GFP and stained with DEUP1 (magenta) and CEP135 (white). Scale bar=5 µm. (**C, D**) Confocal images of control and *Plk4*F/F;R26Cre mTECs expanded by ultrastructure expansion microscopy (UExM). Cells were stained with antibodies against acetylated-α-tubulin

*Figure 5 continued on next page*

*Figure 5 continued*

(white) and GFP (magenta). Blue arrows indicate parent centrioles. Scale bar=0.5 μM. (**E**) Confocal images of control and *Plk4^F/F^*;R26^Cre^ mTECs. Cells were expressing Centrin-GFP and stained with antibodies against γ-tubulin (white) and Centrobin (magenta). Scale bar=5 μm. (**F**) Confocal images of control and *Plk4^F/F^*;R26^Cre^ mTECs. Cells were expressing Centrin-GFP and stained with an antibody against CEP164 (magenta). Scale bar=5 μm. (**G**) Table showing which of the centriole proteins form assemblies in *Plk4^F/F^*;R26^Cre^ mTECs. Data information: All data represent the means ± SEM. *p<0.05. ALI, air-liquid interface; mTEC, mouse trachea epithelial cell.

The online version of this article includes the following source data and figure supplement(s) for figure 5:

**Source data 1.** Values for biological replicates and n for the graph in *Figure 5*.

**Figure supplement 1.** Blocking centriole amplification in mTECs results in assemblies of some centriole proteins.

**Figure supplement 1—source data 1.** Values for biological and technical replicates for graphs in *Figure 5—figure supplement 1*.

*supplement 1D, E*). There was a significant impact on cell health at doses of centrinone above 10 μM which prevented us from exploring whether higher doses of centrinone could fully block centriole amplification.

Previous work shows that treatment with 1 μM of centrinone does not perturb centriole amplification in mTECs (*Nanjundappa et al., 2019*). To define if higher doses of centrinone can suppress centriole assembly, we treated differentiating mTECs with 10 μM centrinone. 10 μM centrinone did not change the percentage of multiciliated mTECs at ALI 7, or the number of centrioles made per cell (*Figure 6—figure supplement 2A-C*). However, there was no increase in PLK4 abundance on deuterosomes in mTECs treated with 1–10 μM of centrinone, a hallmark of kinase inhibition in cycling cells (*Figure 6—figure supplement 2D, E*). This suggests that either PLK4 kinase activity is dispensable for centriole amplification in mTECs or centrinone is unable to effectively inhibit PLK4 in this cell type.

Tracheal epithelial cells express high levels of P-glycoprotein (Pgp) efflux pumps (*Florea et al., 2001*). Given that Pgp efflux pumps confer multidrug resistance, it is plausible that centrinone is actively pumped out of mTECs. To test this, we treated mTECs with 10 μM of centrinone in the presence or absence of the broad-spectrum efflux pump inhibitor Verapamil (*Bellamy, 1996*). While Verapamil alone reduced the percentage of multiciliated mTECs at ALI 7, the number of centrioles produced in multicliated cells was unchanged (*Figure 6—figure supplement 2F, G*). Treatment with 10 μM centrinone and Verapamil reduced the number of centrioles in MCCs to approximately half that observed in cells treated with 10 μM of centrinone alone (*Figure 6—figure supplement 2F, G*). This suggests that drug efflux pumps may serve to reduce the effective intracellular concentration of centrinone in mTECs.

## PLK4 kinase activity is required for centriole assembly in MCCs

Our experiments with centrinone revealed limitations of using small molecule inhibitors to probe the role of PLK4 kinase activity in MCCs. To overcome these drawbacks, we created a conditional kinase-dead allele of PLK4 in mice (*Plk4^cKD^*). The *Plk4^cKD^* allele was generated by knockin of a D154A kinase-dead mutation into exon 5. A *LoxP*-STOP-*LoxP* cassette was inserted into intron 4 of the *Plk4* gene to ensure that the *Plk4^cKD^* allele was only expressed following Cre-mediated recombination (*Figure 6—figure supplement 3A*). As a control, we also created an analogous conditional wild-type allele of *Plk4* that lacked the kinase inactivating mutation (*Plk4^cWT^*) (*Figure 6—figure supplement 3B*).

To determine if PLK4 kinase activity is essential for multiciliogenesis in mTECs, we isolated mTECs from *Plk4^cWT/F^*;R26^Cre^ and *Plk4^cKD/F^*;R26^Cre^ animals and treated them with 4OHT. Following Cre expression, the *Plk4* floxed allele is inactivated and the conditional wild-type or kinase-dead allele is expressed. PLK4 abundance on deuterosomes was ~3.5-fold higher in *Plk4^cKD/F^*;R26^Cre^ mTECs (*Figure 6A and B*). While 4OHT treatment had no impact on centriole amplification in *Plk4^cWT/F^*;R26^Cre^ mTECs, *Plk4^cKD/F^*;R26^Cre^ mTECs possessed large Centrin filaments and few procentrioles. Moreover, while 50% of *Plk4^cWT/F^*;R26^Cre^ mTECs nucleated multicilia, less than 5% of *Plk4^cKD/F^*;R26^Cre^ mTECs contained basal bodies and nucleated multicilia by ALI 7 (*Figure 6C and D*). These data show that PLK4 kinase activity is required to drive centriole biogenesis in mTECs.

To explore the requirement for PLK4 kinase activity in ependymal cells, we infected *Plk4^cWT/F^* and *Plk4^cKD/F^* cells with AAV-Cre. PLK4 was present on the deuterosomes of almost all Cre+ *Plk4^cKD/F^* ependymal cells (*Figure 6—figure supplement 4A*). Moreover, PLK4 abundance on deuterosomes

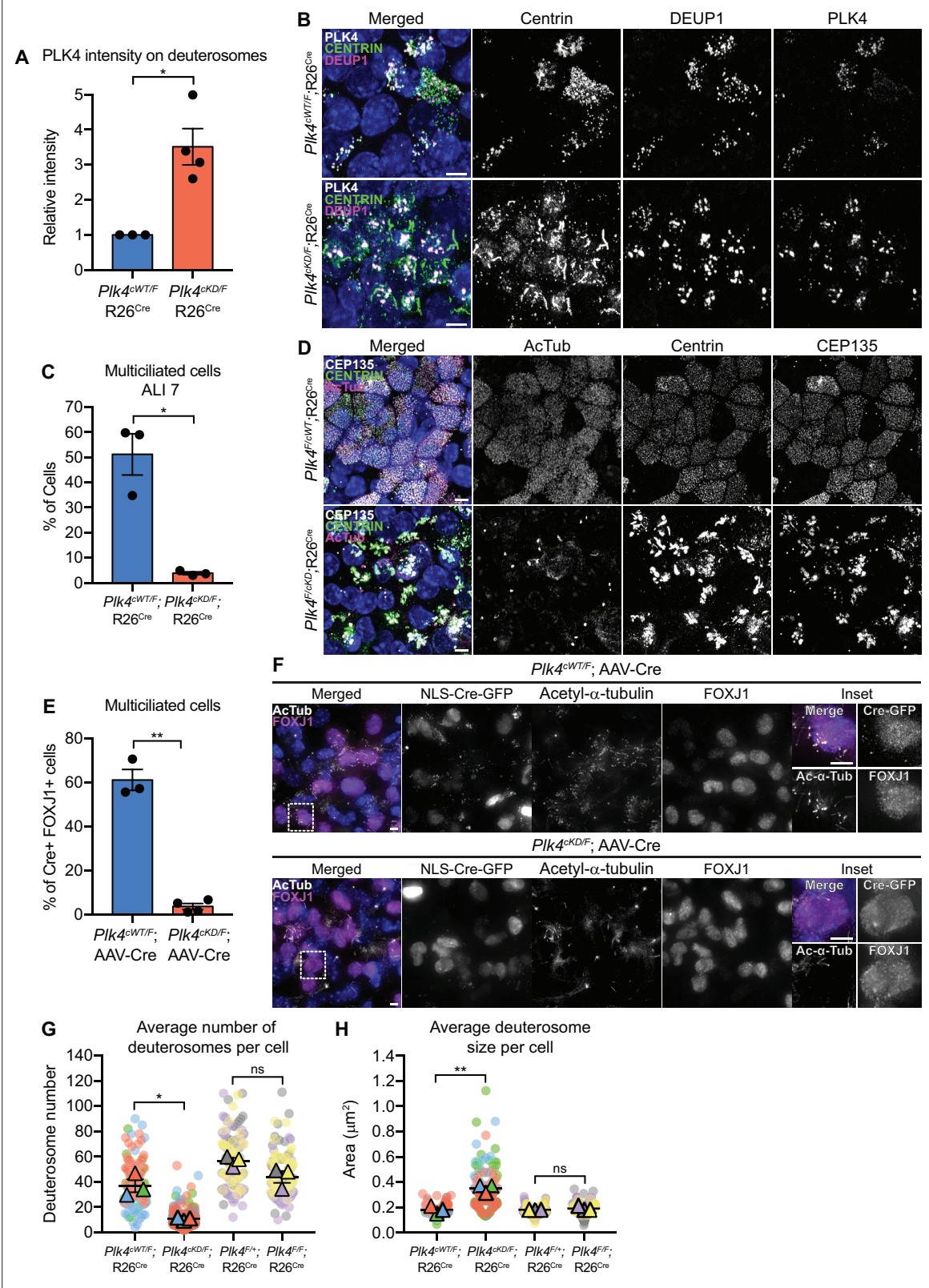

**Figure 6.** Plk4 kinase activity is critical for centriole amplification in MCCs. (**A**) Quantification of the intensity of PLK4 on deuterosomes in mTECs at ALI 2. N=3 (*Plk4^cWT/F^;R26^Cre^*) or 4 (*Plk4^cKD/F^;R26^Cre^*), n=158 (*Plk4^cWT/F^;R26^Cre^*) or 187 (*Plk4^cKD/F^;R26^Cre^*). Two-tailed Welch's t-test. (**B**) Representative confocal images of *Plk4^cWT/F^;R26^Cre^* or *Plk4^cKD/F^;R26^Cre^* mTECs at ALI 2 stained with antibodies against PLK4 (white), Centrin (green), and DEUP1 (magenta). Scale bar=5 μm. (**C**) Quantification of the percent of multiciliated *Plk4^cWT/F^;R26^Cre^* or *Plk4^cKD/F^;R26^Cre^* mTECs at ALI 7. N=3, n>200. Two-tailed Welch's t-test.

*Figure 6 continued*

(**D**) Representative confocal images of *Plk4^cWT/F^*;R26^Cre^ or *Plk4^cKD/F^*;R26^Cre^ mTECs at ALI 7 stained with antibodies against CEP135 (white), Centrin (green), and acetylated-α-tubulin. Scale bar=5 μm. (**E**) Quantification of the percent of FOXJ1+*Plk4^cWT/F^* or *Plk4^cKD/F^* ependymal cells transduced with AAV-Cre-GFP that have multiple cilia at differentiation day 12. N=3, n≥100. Two-tailed Welch's t-test. (**F**) Confocal images of *Plk4^cWT/F^* or *Plk4^cKD/F^* ependymal cells transduced with AAV-Cre-GFP at differentiation day 12. Cells were stained with antibodies against FOXJ1 (magenta) and acetylated-α-tubulin (white). Scale bar=5μm. (**G**) Quantification of the average number of deuterosomes per cell in *Plk4^cWT/F^*;R26^Cre^, *Plk4^cKD/F^*;R26^Cre^, *Plk4^F/+^*;R26^Cre^, or *Plk4^F/F^*;R26^Cre^ mTECs. N=3, n≥50. Two-tailed Welch's t-test. (**H**) Quantification of the average deuterosome size per cell *Plk4^cWT/F^*;R26^Cre^, *Plk4^cKD/F^*;R26^Cre^, *Plk4^F/+^*;R26^Cre^, or *Plk4^F/F^*;R26^Cre^ mTECs. N=3, n≥50. Two-tailed Welch's t-test. Data information: All data represent the means ± SEM. *p<0.05; **<0.01; and not significant indicates p>0.05. MCC, multiciliated cell.

The online version of this article includes the following source data and figure supplement(s) for figure 6:

**Source data 1.** Values for biological and technical replicates for graphs in *Figure 6*.

**Figure supplement 1.** Inhibition of PLK4 kinase activity with centrinone delays centriole amplification in ependymal cells.

**Figure supplement 1—source data 1.** Values for biological and technical replicates for graphs in *Figure 6—figure supplement 1*.

**Figure supplement 2.** Inhibition of PLK4 kinase activity with centrinone does not perturb centriole amplification in mTECs.

**Figure supplement 2—source data 1.** Values for biological and technical replicates for graphs in *Figure 6—figure supplement 2*.

**Figure supplement 3.** A mouse model for the conditional expression of kinase-dead PLK4.

**Figure supplement 4.** PLK4 kinase activity is required for centriole amplification in ependymal cells.

**Figure supplement 4—source data 1.** Values for biological and technical replicates for graphs in *Figure 6—figure supplement 4*.

was ~3-fold higher in Cre+ *Plk4^cKD/F^* ependymal cells (*Figure 6—figure supplement 4B, C*), and STIL staining was absent from deuterosomes (*Figure 6—figure supplement 4D, E*). This is consistent with observations in cycling cells that showed PLK4 kinase activity is required to recruit STIL to the site of procentriole assembly on parent centrioles (*Moyer et al., 2015*). About 60% of Cre+ *Plk4^cWT/F^* ependymal cells formed multicilia. However, Cre+ *Plk4^cKD/F^* ependymal cells failed to undergo centriole amplification and produced very few MCCs (*Figure 6E and F*).

We noted that differentiating mTECs and ependymal cells expressing kinase-dead PLK4 had irregular-shaped deuterosomes that were larger but fewer in number (*Figure 6G and H*, *Figure 6—figure supplement 4F, G*). In contrast, deuterosome size and number were unchanged in differentiating mTECs and ependymal cells that lacked PLK4 (*Figure 6G and H*, *Figure 6—figure supplement 4F, G*). These data suggest that the accumulation of kinase-inactive PLK4 on the surface of deuterosomes changes the properties of these organelles.

## Centriole amplification is required for apical surface area expansion in mTECs

Previous work has shown that centriole number scales proportionally with the apical surface area of muticiliated *Xenopus* epidermal cells and mTECs (*Nanjundappa et al., 2019*; *Kulkarni et al., 2021*). To investigate what drives this scaling phenomenon, we examined apical surface area in muticiliated cells that fail to undergo centriole amplification. The surface area of control mTECs progressively increased throughout differentiation as procentrioles are seeded, elongated, and disengaged from deuterosomes (*Figure 7A and B*).

We next examined apical surface area in PLK4 or SAS4 knockout mTECs that failed centriole amplification and contained filamentous assemblies of Centrin. As a control, we also quantified cells in the same culture that escaped Cre-mediated recombination and underwent centriole amplification. At ALI 3, when most muticiliated cells are in the early stages of differentiation, the average apical surface area was similar in control and PLK4 knockout mTECs (*Figure 7C and D*). However, while control mTECs increased their average apical surface area from 90 to 159 μm² from ALI 3 to ALI 7, PLK4 and SAS4 knockout mTECs failed to undergo this expansion. Notably, apical surface area expansion was still observed in MBB phase cells that escaped Cre-recombination (*Figure 7C, E and F*). These data show that centriole amplification plays a critical role in promoting apical membrane expansion in differentiating mTECs.

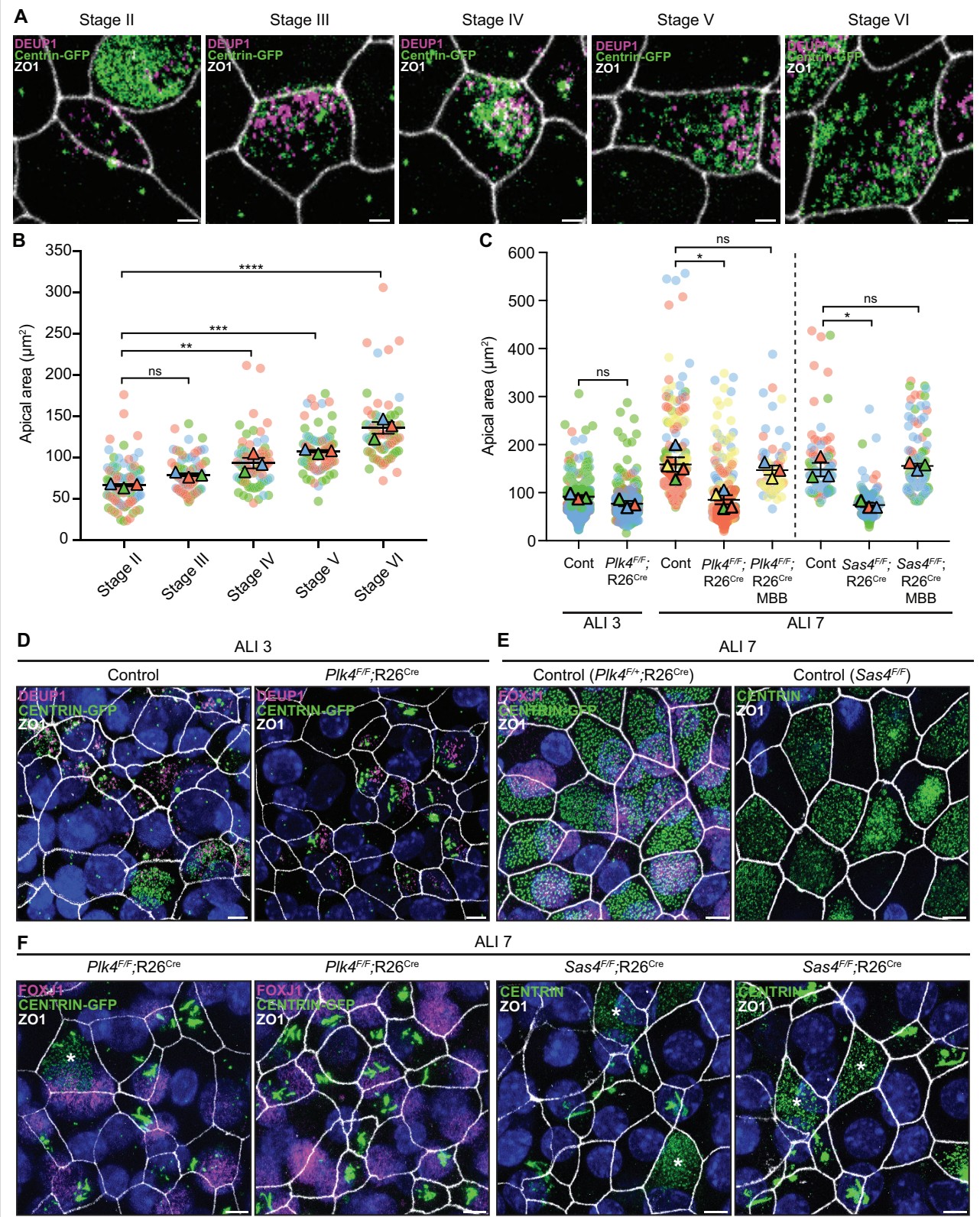

**Figure 7.** Centriole amplification promotes apical surface area expansion in mTECs. (**A**) Representative confocal images of control mTECs in stages II-VI of centriole amplification. Cells were expressing Centrin-GFP and stained with antibodies against DEUP1 (magenta) and ZO1 (white). Scale bar = 2 μm. (**B**) Graph of the apical area of control mTECs (marked with ZO1) at ALI 3 at different stages of centriole amplification. Circles represent individual cells and triangles represent the average per biological replicate. N=3, n≥20. One-way ANOVA with *post hoc* analysis. (**C**) Graph of the apical area of control,

*Figure 7 continued on next page*

*Figure 7 continued*

*Plk4^F/F*;R26^Cre and *Sas4^F/F*;R26^Cre mTECs at ALI 3 and ALI 7. The area across all stages of centriole amplification was pooled per condition. MBB phase cells that presumably escaped Cre-mediated recombination serve as an internal control. Circles represent individual cells and triangles represent the average per biological replicate. N≥3, n≥50. Two-tailed Welch's t-test (ALI 3). One-way ANOVA with *post hoc* analysis (ALI 7). (**D**) Representative confocal images of control and *Plk4^F/F*;R26^Cre mTECs at ALI 3. Cells were expressing Centrin-GFP and stained with antibodies against DEUP1 (magenta) and ZO1 (white). Scale bar = 5 μm. (**E**) Representative confocal images of control mTECs at ALI 7. *Plk4^F/+*;R26^Cre cells were expressing Centrin-GFP and stained with antibodies against FOXJ1 (magenta) and ZO1 (white). *Sas4^F/F* cells were stained with antibodies against Centrin (green) and ZO1 (white). Scale bar = 5 μm. (**F**) Representative confocal images of *Plk4^F/F*;R26^Cre and *Sas4^F/F*;R26^Cre mTECs at ALI 7. *Plk4^F/F*;R26^Cre cells were expressing Centrin-GFP and stained with antibodies against FOXJ1 (magenta) and ZO1 (white). *Sas4^F/F*;R26^Cre cells were stained with antibodies against Centrin (green) and ZO1 (white). MBB phase cells that presumably escaped Cre-mediated recombination are marked with an *. Scale bar = 5 μm. Data information: All data represent the means ± SEM. *P < 0.05; **< 0.01; ***< 0.001; ****< 0.0001 and not significant indicates P > 0.05.

The online version of this article includes the following source data for figure 7:

**Source data 1.** Values for biological and technical replicates for graphs in *Figure 7*.

## Discussion

MCCs transcriptionally upregulate many of the proteins required for centriole duplication in cycling cells, but whether the molecular mechanisms driving centriole assembly are the same in cycling cells and MCCs remains unclear (*Hoh et al., 2012*; *Revinski et al., 2018*; *Zaragosi et al., 2020*). In this manuscript, we employed a genetic approach to test the impact of inactivating the *Plk4* gene or kinase activity in differentiating mouse MCCs. Conditional knockout of PLK4 blocked procentriole assembly and multiciliogenesis without preventing MCC differentiation. Although the PLK4 receptor CEP152 was recruited to deuterosomes and parent centrioles in MCCs lacking PLK4, the cartwheel proteins STIL and SAS6 were not, demonstrating that PLK4 plays a crucial role in the earliest stages of centriole amplification in MCCs. These results conflict with prior work that observed a delay in centriole assembly in MCCs that were depleted of PLK4 using an shRNA (*Nanjundappa et al., 2019*). The most likely explanation for this discrepancy is that shRNA-mediated depletion of PLK4 failed to achieve a sufficient level of target depletion required to block centriole assembly. Taken together, our data show the centriolar and deuterosomal pathways of centriole biogenesis in MCCs have a striking resemblance to the established centriole assembly pathway of cycling cells.

While high doses of the PLK4 inhibitor centrinone reduced centriole production in ependymal MCCs, this was not the case in mTECs. Similarly, a previous study also reported that there was no decrease in centriole number in mTECs treated with centrinone (*Nanjundappa et al., 2019*). These data could be interpreted to suggest that PLK4 kinase activity is dispensable for centriole amplification, at least in mTECs (*Nanjundappa et al., 2019*). However, the doses of centrinone used may not fully inhibit PLK4 activity in mTECs, possibly because of the expression of multidrug-resistant transporters that pump drugs out of tracheal epithelial cells (*Florea et al., 2001*). To address this limitation, we developed a strategy to conditionally ablate PLK4 kinase activity using genetic tools that enable specific and more complete PLK4 inhibition. We show that selectively ablating PLK4 kinase activity leads to a failure of centriole amplification in mTECs and ependymal cells, thereby phenocopying the loss of PLK4 protein. MCCs lacking PLK4 kinase activity possessed fewer deuterosomes that were larger in size. Since deuterosomes are unaffected in MCCs lacking PLK4, we speculate that deuterosomes loaded with kinase-inactive PLK4 undergo fusion events that are driven by the self-assembly properties of PLK4 (*Montenegro Gouveia et al., 2018*; *Yamamoto and Kitagawa, 2018*; *Park et al., 2019*).

Centriole amplification in MCCs occurs within a cloud of PCM and fibrogranular material. Several centriole components are concentrated within the fibrogranular material, and this has been proposed to facilitate the rapid assembly of centrioles in MCCs (*Mercey et al., 2019*; *Zhao et al., 2021*). In the absence of PLK4, or the centriole protein SAS4, multiple centriole, PCM, and distal appendage proteins formed large independent assemblies in mTECs. These assemblies were not observed in multiciliated ependymal cells, which produce ~3 times fewer centrioles than mTECs and likely have lower concentrations of centriole proteins. The high abundance of centriole proteins in differentiating mTECs is likely to drive self-interactions that promote the formation of distinct assemblies when centriole amplification is blocked.

In addition to containing aggregates of centriole proteins, mTECs that fail centriole amplification have a smaller apical surface area. Centriole number has been shown to scale with apical area in

multiciliated mTECs and *Xenopus* epithelial cells (*Nanjundappa et al., 2019*; *Kulkarni et al., 2021*). Yet, it remains unclear exactly how this scaling relationship is established. Our results show that the apical area of mTECs progressively expands during differentiation, but this fails to occur in cells that lack PLK4 or SAS4 and do not undergo centriole amplification. This suggests that centriole amplification is a central driver of surface area expansion in mammalian MCCs. In *Xenopus* epithelial cells, depleting centriole number did not reduce the surface area of MCCs (*Kulkarni et al., 2021*). However, multiciliated *Xenopus* epithelial cells undergo two waves of centriole amplification that take place before and after radial intercalation (*Kulkarni et al., 2021*). By contrast, mammalian MCCs do not radially intercalate and use a single round of centriole production (*Al Jord et al., 2014*). This suggests that multiple pathways could operate to calibrate centriole numbers with the MCC apical area. Future studies will be required to define how centriole amplification feeds back to tune the apical surface area of mammalian MCCs.

# Materials and methods

**Key resources table**

| Reagent type (species) or resource | Designation | Source or reference | Identifiers | Additional information |
|---|---|---|---|---|
| Antibody | Rabbit polyclonal CEP164 | EMD Millipore | ABE2621 | (IF) use 1:1000 |
| Antibody | Mouse monoclonal acetylated-α-tubulin | Cell Signaling Technology | 12152 | (IF) use 1:1000, (IHC) use 1:500 |
| Antibody | Rat polyclonal ZO-1 | Thermo Fisher Scientific | 14-9776-82 | (IF) use 1:1000 |
| Antibody | Rabbit polyclonal DEUP1 | *Mercey et al., 2019*. https://doi.org/10.1038/s41556-019-0427-x | | (IF) use 1:1000 |
| Antibody | Mouse monoclonal SAS6 | Santa Cruz Biotechnology | sc-81431 | (IF) use 1:1000 |
| Antibody | Rabbit polyclonal PCNT | Abcam | ab4448 | (IF) use 1:1000 |
| Antibody | Mouse monoclonal FOXJ1 | Thermo Fisher Scientific | 14-9965-82 | (IF) use 1:1000 |
| Antibody | Rabbit polyclonal PLK4 | *Moyer et al., 2015*. https://doi.org/10.1083/jcb.201502088 | | (IF) use 1:1000 |
| Antibody | Rabbit polyclonal STIL | *Moyer et al., 2015*.https://doi.org/10.1083/jcb.201502088 | | (IF) use 1:1000 |
| Antibody | Rabbit polyclonal CEP135 | This study | | Homemade, raised against a.a. 649–1140. (IF) use 1:1000 |
| Antibody | Rabbit polyclonal acetylated-α-Tubulin | Cell Signaling Technology | 5335 | (IF) use 1:1000, (IHC) use 1:500 |
| Antibody | Rabbit polyclonal CNTROB | Atlas Antibodies | HPA023321 | (IF) use 1:1000 |
| Antibody | Rabbit polyclonal CEP152 | This study | | Homemade, raised against a.a. 491–810. (IF) use 1:1000 |
| Antibody | Rabbit polyclonal CDK5RAP2 | Bethyl | BETIHC-00063 | (IF) use 1:1000 |
| Antibody | Goat polyclonal CEP192 | *Moyer and Holland, 2019*. https://doi.org/10.7554/eLife.46054 | | (IF) use 1:1000 |
| Antibody | Goat polyclonal g-Tubulin | *Levine et al., 2017*. https://doi.org//10.1016/j.devcel.2016.12.022 | | (IF) use 1:1000 |
| Chemical compound, drug | Centrinone | Tocris Bioscience | Cat. no. 5687 | Varying concentrations |
| Chemical compound, drug | Verapamil | Cytoskeleton, Inc | Cat. #CY-SC002 | Use at 10 µM |

*Continued on next page*

*Continued*

| Reagent type (species) or resource | Designation | Source or reference | Identifiers | Additional information |
|---|---|---|---|---|
| Other | DAPI Stain | MilliporeSigma | Cat. no. 10236276001 | (IF) use 1:1000 |

## Animals

Mice were housed and cared for in an AAALAC-accredited facility. All animal experiments were approved by the Johns Hopkins University Institute Animal Care and Use Committee (MO21M300). All studies employed a mixture of male and female mice and no differences between sexes were observed.

## Generation of *Plk4*^F/F mice

*Plk4*^F/F mice were generated using the *Easi*-CRISPR method (**Quadros et al., 2017**). This strain will be available at The Jackson Laboratory Repository (JAX Stock no. 037549). A 1420 base pair single-stranded DNA donor template was injected along with pre-assembled crRNA + tracrRNA + Cas9 ribonucleoprotein (ctRNP) complexes into B6SJLF1/J mouse zygotes (JAX stock #100012). Microinjection was performed by the Johns Hopkins Transgenic Core Laboratory. The single-stranded DNA donor template and crRNA guides were synthesized by IDT. The crRNAs targeted the following DNA sequences: 5′-aagctaggacttaatactc-3′ and 5′- ctgcatgtagagggaagctg-3′. The crRNA sequences were: /AltR1/rArA rGrCrU rArGrG rArCrU rUrUrA rArUrA rCrUrC rGrUrU rUrUrA rGrArG rCrUrA rUrGrC rU/AltR2/; and: /AltR1/rCrU rGrCrA rUrGrU rArGrA rGrGrG rArArG rCrUrG rGrUrU rUrUrA rGrArG rCrUrA rUrGrC rU/AltR2/. The 1420-bp donor template used was: 5′-tacaattaggcaataaaactactgttcat atcagaaggcagtcagtatcacagctattcctcactcagccccagataacttcgtataatgtatgctatacgaagttatcttccctctacat gcagctattagtcctttagcgatcaaacttgttcaataagtttaaaagattatccttatactgctcagtcattatacggtctttcaaaagatcctt aactattggctgtttggtgtttaatgaagttaaaacattgacattgcttttggtgggtgaatacctgtgttgattttcatccagtgatcttctagg ctttgaaagcatttctactgagtgacaacgttctactgagtatgttttttgctcgagactgattagaggggctggctctatcagaggaatgag ctctgcgcaggtatcgagaatgcggcctctcttctgcatcttcaatcactctcccccgtccctactattagcttcttgttctggatttccccatt gagtacaaaaattacttccatctcctgaagaaaagtcacttgaatttttatttttttgaaatacagtaattttatttggaagtggttgaccaacca aaagtcttctgtcaagtaggctgccactcaaactggtaccagaagaggctgtaattgttgtggaaagtgtagcatgcccactatccattgag tcctctacagtccctacgtctttactctttggtgaaggatttcgtgacatgaaaggatggtccaacacagaagacagacttaaccgatctgc agggtttctacgaagtaactggtggataaggtcctgggcctctcgtgacaaaaaggctggcatttcataatctgccaggactactttg ttcaatgtgttcttgactgtgtcagtgtcaaaaggtggtcttccaataagtaacgtataaaacatacagcccaatgaccaaatatcagattca agtccatgtgcacttcgagttgcaatttctggtgaaatataattaggagtcccacagagtgtatagtgcttttcatgtggcatattcaactg cgttgctagtccaaagtcagcaatttttatgttcatattccgcgtaagtaagatgttagagagtgtgaggtcccggtgcaatatgccatgag aatgaagatataacattcctgtgataatctggtcatgaagtgcctagctgtgaagatagaaaaagagagagcttaacaattcaaatc tgggaaatttattgtttagaatacacgataaaatctagtatacacgccagagataacttcgtataatgtatgctatacgaagttattattaa agtcctagcttgcaggttagatttatcaactaaaacaccattctatcaatataatgctatctaattcccctcaagaaaactagacactccttt-3′. Genotyping was performed with the following primers: Common Fwd: 5′-tcttgaggggaattagatagca-3′, WT Rev: 5′-tgcaatatgccatgagaatga-3′, and LoxP Rev: 5′-ctcactcagccccagataac-3′.

## Generation of *Plk4*^cWT and *Plk4*^cKD mice

Mice were created using CRISPR-Cas9 genome editing. A double-stranded DNA donor template was injected along with pre-assembled crRNA + tracrRNA + Cas9 ribonucleoprotein (ctRNP) complexes into B6SJLF1/J mouse zygotes (JAX stock #100012). Microinjection was performed by the Johns Hopkins Transgenic Core Laboratory. The donor template was amplified by PCR from plasmid DNA containing a *LoxP*-STOP-*LoxP* cassette flanked by a 1983-bp 5′ homology arm and a 1445-bp 3′ homology arm. The crRNA guide was synthesized by IDT and targeted the following DNA sequence: 5′-aagctaggacttaatactc-3′.

*Centrin-GFP mice. ROSA-Neo-EGFP-Cetn1* mice. Gt(ROSA)26Sortm1(EGFP/CETN1)Sev/J were obtained from the Jackson laboratory **Hirai et al., 2016** (strain #029363). Mice were crossed to a deleter strain to generate constitutive Centrin-GFP expression in all tissues (Centrin-GFP mice). Genotyping was performed with the following primers: Centrin-GFP 1: 5′-AAAGTCGCTCTGAGTTGTTAT-3′,

Centrin-GFP 2: 5′-GCGAAGAGTTTGTCCTCAACC-3′, Centrin-GFP 3: 5′-GGAGCGGGAGAAATGGA
TATG-3′.

### Rosa26<sup>CreERT2</sup> mice

Rosa26<sup>CreERT2</sup> mice (B6.129-Gt(ROSA)26Sortm1(cre/ERT2)Tyj/J) were obtained from the Jackson labo-
ratory (**Ventura et al., 2007**) (strain #008463). Genotyping was performed with the following primers:
Reaction 1: WT Fwd: 5′-CTGGCTTCTGAGGACCG-3′, WT Rev: 5′-CCGAAAATCTGTGGGAAGTC-
3′, Reaction 2: Mut Fwd: 5′-CGTGATCTGCAACTCCAGTC-3′, Mut Rev: 5′-AGGCAAATTTTGGTGTA
CGG-3′.

### Nestin<sup>Cre</sup> mice

Nestin<sup>Cre</sup> mice were obtained from The Jackson Laboratory (stock #003771) (**Tronche et al., 1999**).
The following primers were used for genotyping: Nestin<sup>Cre</sup> Fwd 5′-ATTGCTGTCACTTGGTCGTGGC-3
′; Nestin<sup>Cre</sup> Rev 5′-GGAAAATGCTTCTGTCCGTTTGC-3′.

### *Usp28*<sup>F/F</sup> mice

*Usp28*<sup>F/−</sup> mice were obtained from the laboratory M. Eilers (University of Wuerzburg, Germany)
(**Diefenbacher et al., 2014**). The following primers were used for genotyping: Usp28 Common Fwd
1718 27 5′-GAGGCTTGAGTTATGACTGG-3′; Usp28 WT Rev 1718 m28 5′-AGAACACCTGCTGCTTA
AGC-3′; Usp28 Del Rev 5′- TCCCCCAAGAGTGTTTTCAC-3′.

## Cell culture

### Primary cells

mTEC cultures were harvested and grown as previously described (**You and Brody, 2013**). Briefly,
tracheas were harvested from mice that were between 3 weeks and 12 months old. Tracheas were
then incubated overnight in Pronase (Roche) at 4°C. Tracheal cells were dissociated by enzymatic and
mechanical digestion the following day. The cells were plated onto 0.4 µm Falcon transwell membranes
(Transwell, Corning). Once the cells were confluent (proliferation day 3), 2.5 µM 4-Hydroxytamoxifen
(4OHT) was added to the apical and basal chambers. On proliferation day 5, the medium from the
apical chamber was removed and the basal medium was replaced with NuSerum medium containing
2.5 µM 4OHT. This time point is considered air-liquid interface day 0 (ALI 0). The basal media were
changed every 2 days and the 4OHT was replenished. Cells were allowed to differentiate until needed
for analysis (ALI days 3, 7, or 21). For cultures analyzed at ALI 21. 4OHT was supplied to the media
until ALI 7. For mTECs cultured in centrinone and/or verapamil, mTEC NS media containing 1 µM,
2.5 µM, 5 µM, 10 µM of centrinone and/or 10 µM Verapamil was added to the basal chamber and
replenished every other day.

For the mouse ependymal cell cultures, brains were dissected from P0 to P2 mice, dissociated
and cultured as previously described (**Delgehyr et al., 2015**). Briefly, the mice were sacrificed by
decapitation. The brains were dissected in Hank's solution (10% HBSS, 5% HEPES, 5% sodium bicar-
bonate, and 1% penicillin/streptomycin (P/S) in pure water) and the telencephalon was manually cut
into pieces, followed by enzymatic digestion (DMEM GlutaMAX 1% P/S, 3% papain [Worthington
3126], 1.5% 10 mg ml<sup>−1</sup> DNAse, and 2.4% 12 mg ml<sup>−1</sup> cystein) for 45 min at 37°C in a water bath. The
digestion was stopped by the addition of a solution of trypsin inhibitors (Leibovitz's L15 medium, 10%
1 mg ml<sup>−1</sup> trypsin inhibitor [Worthington], and 2% 10 mg ml<sup>−1</sup> DNAse). The cells were then washed in
L15 medium and resuspended in DMEM GlutaMAX supplemented with 10% fetal bovine serum (FBS)
and 1% P/S in a poly-l-lysine-coated flask. The ependymal progenitors were allowed to proliferate for
4–5 days, until confluence was reached, before being incubated overnight with shaking (250 r.p.m.),
and then re-plated at a density of $7 \times 10^4$ cells cm<sup>−2</sup> (differentiation day –1) in DMEM GlutaMAX, 10%
FBS and 1% P/S on poly-l-lysine-coated coverslips. The following day, the medium was replaced with
serum-free DMEM GlutaMAX 1% P/S to trigger ependymal differentiation in vitro (differentiation day
0).

### Adeno-associated virus transduction of ependymal cells

At differentiation day 0 ependymal cells were washed 2× with phosphate-buffered saline (PBS).
pAAV.CMV.HI.eGFP-Cre.WPRE.SV40 (Addgene, cat. no. 105545-AAV1) was resuspended in DMEM

GlutaMAX 1% P/S at an MOI of 100,000. About 250 µl of viral solution was added per well. Cells were incubated with virus for 4–5 hr, and then the viral media was replaced with 1 ml of DMEM GlutaMAX with 1% P/S.

## Processing of cells and tissues for immunofluorescence microscopy

### mTEC cultures

Membranes were fixed in 100% ice-cold methanol for 10 min at −20°C. The membranes were washed 3× with PBS for 5 min each. The membranes were then blocked at room temperature (RT) for 1 hr in PBS with 0.2 M Glycine, 2.5% FBS, and 0.1% Triton X-100. The membranes were cut in half and incubated with primary antibodies diluted in blocking buffer for 1 hr at RT. Next, the membranes were washed 3× in PBS for 5 min each and incubated for 45 min at RT with secondary antibodies and DAPI diluted in blocking buffer. The membranes were again washed 3× in PBS for 5 min each and mounted onto slides using ProLong Gold Antifade (Invitrogen). Images of stained mTECs were obtained using an SP8 (Leica Microsystems) confocal microscope. Images were collected using a Leica 63× 1.40 NA oil objective at 0.5 µM z-sections.

### Ependymal cell cultures

Cells were grown on 12 mm glass coverslips as described above and fixed for 10 min in either 4% paraformaldehyde (PFA) (for FOXJ1) at RT or 100% ice-cold methanol at −20°C. The samples were blocked in PBS with 0.2% Triton X-100 and 10% FBS before incubation with the primary and secondary antibodies. The cells were counterstained with DAPI (10 µg ml$^{-1}$; Sigma-Aldrich) and mounted in ProLong Gold Antifade (Invitrogen). Immunofluorescence images were collected using a Deltavision Elite system (GE Healthcare) controlling a Scientific CMOS camera (pco.edge 5.5). Acquisition parameters were controlled by SoftWoRx suite (GE Healthcare). Images were collected at RT using an Olympus 60× 1.42 NA oil objective at 0.2 mM z-sections and subsequently deconvolved in SoftWoRx suite.

### Brains for the ex vivo imaging of ependymal cells

To isolate the lateral wall for immunofluorescence, brains from P21 *Plk4$^{F/F}$;Usp28$^{F/F}$*;Nestin$^{Cre}$ mice and littermate controls were dissected as described previously (*Mirzadeh et al., 2010*). Animals were sacrificed by cervical dislocation and the head was removed. The brain was dissected out of the skull and placed in a petri dish with PBS. The remainder of the dissection was performed under the stereomicroscope. First, the olfactory bulbs were dissected away from the brain. The brain was then divided along the intrahemispheric fissure and a coronally oriented cut was made at the posterior-most aspect of the interhemispheric fissure, allowing the caudal hippocampus to be visualized in cross-section. The hippocampus was then released from the overlying cortex. The lateral wall then completely exposed by removing any overhanging cortex dorsally and the thalamus ventrally. The dissected brains were then fixed overnight in 4% PFA in PBST (PBS + 0.1% Triton X-100) at 4°C. The following day, brains were washed 3× in PBST for 20 min at RT, and then incubated with 30% sucrose in PBS for 24 hr at 4°C. Brains were embedded in O.C.T. medium and frozen for cryosectioning. About 20 µm brain sections were cut using a Leica CM3050 cryostat and collected on Superfrost microscope slides (Thermo Fisher Scientific). For immunohistochemistry staining, the brain sections were rehydrated with PBS for 5 min at RT, blocked and permeabilized with blocking solution (10% donkey serum [Sigma-Aldrich], 0.5% Triton X-100 in PBS) for 1 hr at room temperatureRT and incubated with the respective primary antibodies diluted in blocking solution for 12 hr at 4°C. Following primary antibody staining, the tissue slides were washed three times with PBS + 0.5% Triton X-100 and incubated with secondary antibodies and DAPI diluted in blocking solution for 1 hr at RT. Following secondary antibody staining, the tissue slides were washed three times with PBS + 0.5% Triton X-100 before mounting. Secondary antibodies were conjugated to Alexa Fluor 488, 555, or 647 (Thermo Fisher Scientific, 1:500). Images of stained sections were obtained using an SP8 (Leica Microsystems) confocal microscope. Images were collected using a Leica 63× 1.40 NA oil objective at 0.5 µM z-sections.

### Brains for analysis of telencephalon area and ex vivo imaging of NPCs

For analysis of embryonic brain sections, timed pregnant females carrying the respective transgenes were anesthetized with isoflurane before cervical dislocation. The heads of the embryos were removed

and rinsed with ice-cold PBS, followed by incubation in 4% PFA in PBS overnight at 4°C. The next day, the brains were dissected out of the skulls and rinsed three times with PBS before being imaged using a MU1003 digital camera attached to a dissecting microscope (Amscope). The brains were then incubated in 30% sucrose at 4°C for 24 hr. Brains were embedded in O.C.T medium and frozen for cryosectioning. 20 μm brain sections were cut using a Leica CM3050 cryostat and collected on Super-frost microscope slides (Thermo Fisher Scientific). Tissue staining was performed, and images were collected as described above.

## Ultrastructure expansion microscopy (U-ExM)

### Ependymal cells

U-ExM was carried out as previously described (*Gambarotto et al., 2021*). Coverslips with unfixed cells were incubated in a solution of 0.7% formaldehyde with 1% acrylamide in PBS for 5 hr at 37°C. Gelation was carried out via incubation of coverslips with cells facing down with 35 μl of U-ExM MS composed of 19% (wt/wt) sodium acrylate, 10% (wt/wt) acrylamide, 0.1% (wt/wt) N,N'-methylenbi-sacrylamide (BIS) in PBS supplemented with 0.5% APS and 0.5% TEMED, on Parafilm in a pre-cooled humid chamber. Gelation proceeded for 10 min on ice, and then samples were incubated at 37°C in the dark for 1 hr. A 4 mm biopsy puncher (Integra; 33–34 P/25) was used to create one punch per coverslip. Punches were transferred into 1.5 ml Eppendorf tubes containing 1 ml denaturation buffer (200 mM SDS, 200 mM NaCl, and 50 mM Tris in ultrapure water, pH 9) and incubated at 95°C for 1 hr. After denaturation, gels were placed in beakers filled with ddH2O for the first expansion. Water was exchanged at least two times every 30 min at RT, and then gels were incubated overnight in ddH2O. Next, to remove excess water before incubation with primary antibody solution, gels were placed in PBS two times for 15 min. Incubation with primary antibody diluted in 2% PBS/BSA was carried out at 37°C for 2.5 hr, with gentle shaking. Gels were then washed in PBST three times for 10 min with shaking and subsequently incubated with secondary antibody solution diluted in 2% PBS/BSA for 2.5 hr at 37°C with gentle shaking. Gels were then washed in PBST three times for 10 min with shaking and finally placed in beakers filled with ddH2O for expansion. Water was exchanged at least two times every 30 min, and then gels were incubated in ddH2O overnight. Gel expanded between 4.0× and 4.5× according to SA purity.

### mTEC cultures

U-ExM was carried out as previously described with a few modifications (*Gambarotto et al., 2021*; *Kong and Loncarek, 2021*). Transwell filters containing mTECs were fixed in freshly made 4% PFA in PBS for 1 hr at RT. Filters were then incubated in a solution of 30% acrylamide with 4% formaldehyde at 37°C for 16 hr. Filters were washed 3× with PBS for 10 min at RT. Filters were cut out of their tran-swell holders and placed cell-side up on a parafilm-coated petri dish on ice. After cooling for 20 min, complete polymerization mixture was added to the filters (20% acrylamide, 0.04% bisacrylamide, 7% sodium acrylate, 0.5% TEMED, 0.5% ammonium persulfate, in 1× PBS) and a coverslip was placed on top. Samples were kept on ice for 20 min, and then incubated at RT for 1 hr. One punch was cut from each filter using a 4-mm biopsy puncher. Punches were boiled in SDS buffer (0.2 M NaCl, 0.05 M Tris-HCl, and 5% SDS) for 1 hr. Punches were washed 10× 20 min in PBS at RT. Immunolabeling and expansion of samples was performed as described above.

## Quantification of centriole number in fixed samples

### Ependymal cells

Regions that contained a high density of MCCs were selected. At least three fields of view were selected per sample. In *Plk4*^F/F^; AAV-Cre ependymal cell samples, all DEUP1+ cells within those fields of view were counted. In control ependymal cell samples, all MCCs in the disengagement phase of centriole amplification were counted. Cells for which individual centrioles could not be resolved were excluded from our analysis. The CEP135 foci in each cell were quantified using a semi-automated ImageJ macro. A circle was drawn around all CEP135 foci in each cell. A threshold was applied to each image, and CEP135 intensity maxima were identified and counted by ImageJ.

### mTEC cultures

Measurements of apical area were made using ImageJ. A perimeter was drawn around the apical area of individual cells by hand using the polygon selection tool and ZO1 as a marker, and the measurement function of ImageJ was used to measure the apical area. Centriole number per cell was quantified as described above.

## Quantifications of fluorescence intensities

Regions that contained a high density of MCCs were selected. At least three fields of view were selected per sample and all DEUP1+ cells were analyzed. PLK4 intensity on deuterosome measurements were made using a semi-automated ImageJ macro. A circle was drawn to encompass all deuterosomes for a given cell. Deuterosomes were identified by DEUP1 signal based on a pre-set threshold which was kept the same throughout all images in a replicate. Signal intensity of PLK4 (raw integrated density) was then measured within the DEUP1 threshold. For a given cell, the background-subtracted intensity within each thresholded region was added together to calculate the total PLK4 intensity on deuterosomes per cell. The SAS6 and STIL intensity measurements were collected in the same way.

## Quantifications of deuterosome number and area

Quantification of deuterosome number and area was performed using a semi-automated ImageJ macro. Cells were manually segmented and deuterosomes were identified by automated thresholding of the DEUP1 signal. DEUP1 intensity maxima were used to count deuterosome numbers. The area of the thresholded region was divided by the deuterosome number to calculate the average deuterosome area.

## Fluorescence recovery after photobleaching

To prepare cells for photobleaching experiments, WT and PLK4 KO mTECs expressing GFP-Centrin were isolated and differentiated as described above. On ALI day 3, the filters with adhered cells were cut out from the transwell insert and inverted onto a glass coverslip with a drop of differentiation media. A plastic ring was placed on top of the filter to keep it in contact with the coverslip. Cells were imaged on a Leica Sp8 inverted microscope using a 63×/1.4 NA objective. An region of interest (ROI) was drawn around all or part of a GFP-Centrin signal at the centrioles or aggregates and photobleached to approximately 80% of the pre-bleach fluorescence intensity with 10 pulses of 488 nm laser light (1039 ms exposure). A Z-series of 13 planes (0.5 µm step size) was acquired immediately before and after bleaching. GFP-Centrin fluorescence intensities within the bleached ROI were measured using ImageJ. The first post-bleach intensity was subtracted from the pre-bleach and post-bleach values and then normalized to give the percentage of the pre-bleach intensity.

## Statistical analysis

Statistical analysis was performed using GraphPad Prism software. Differences between biological replicates (N) were analyzed using a two-tailed unpaired Student's t-test (Welch's t-test) or one-way ANOVA with post hoc analysis as indicated in the figure legends. Error bars represent SEM. Figure legends state the number of animals/biological replicates (N) and cells (n) per experiment.

## Original figures

*Figures 1A and 3A* were created with https://biorender.com/.

## Acknowledgements

This study was supported by the National Institutes of Health Grants R01GM114119, R01GM133897, and R01CA266199 (to AJH).

## Additional information

### Funding

| Funder | Grant reference number | Author |
|---|---|---|
| National Institute of General Medical Sciences | R01GM114119 | Andrew Jon Holland |
| National Institute of General Medical Sciences | R01GM133897 | Andrew Jon Holland |
| National Cancer Institute | R01CA266199 | Andrew Jon Holland |

The funders had no role in study design, data collection and interpretation, or the decision to submit the work for publication.

### Author contributions

Gina M LoMastro, Conceptualization, Data curation, Formal analysis, Investigation, Writing - original draft, Writing - review and editing; Chelsea G Drown, Aubrey L Maryniak, Formal analysis, Investigation; Cayla E Jewett, Margaret A Strong, Data curation, Investigation; Andrew Jon Holland, Formal analysis, Supervision, Funding acquisition, Writing - original draft, Project administration, Writing - review and editing

### Author ORCIDs

Gina M LoMastro (ID) http://orcid.org/0000-0001-9751-1146
Cayla E Jewett (ID) http://orcid.org/0000-0002-8406-0814
Andrew Jon Holland (ID) http://orcid.org/0000-0003-3728-6367

### Ethics

Mice were housed and cared for in an AAALAC-accredited facility. All animal experiments were approved by the Johns Hopkins University Institute Animal Care and Use Committee (MO21M300). All studies employed a mixture of male and female mice and no differences between sexes were observed. Euthanasia was performed using isoflurane followed by cervical dislocation and every effort was made to minimize animal suffering.

### Decision letter and Author response

Decision letter https://doi.org/10.7554/eLife.80643.sa1
Author response https://doi.org/10.7554/eLife.80643.sa2

## Additional files

### Supplementary files

• MDAR checklist

### Data availability

All data generated or analysed during this study are included in the source data file.

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
