## [Editor Report]

PLK4 is the master regulator of centriole biogenesis, but whether it is also key for centriole amplification during differentiation of multiciliated cells (MCCs) has been questioned based on PLK4 chemical inhibition. In this fundamental study, using mouse models engineered to lack PLK4 or PLK4 activity, LoMastro et al., provide exceptional evidence that PLK4 and its activity are essential for centriole amplification in MCCs. Moreover, they show that centriole amplification in MCCs drives expansion of their apical surface. The findings settle the debate whether PLK4 is crucial for centriole amplification in multiciliated cells and will determine how cell biologists and experts interested in multi-ciliogenesis-related pathologies understand this process.

---

## [Decision Letter]

**Decision letter after peer review:**

Thank you for submitting your article "PLK4 drives centriole amplification and apical surface area expansion in multiciliated cells" for consideration by *eLife*. Your article has been reviewed by 2 peer reviewers, one of whom is a member of our Board of Reviewing Editors, and the evaluation has been overseen by Anna Akhmanova as the Senior Editor. The following individual involved in review of your submission has agreed to reveal their identity: Francesc R Garcia-Gonzalo (Reviewer #2).

Essential revisions:

Both reviewers are very enthusiastic about your study. Before publication, please revise your manuscript addressing a few points raised by the reviewers. These only require some changes in text and figures.

*Reviewer #1 (Recommendations for the authors):*

This is a very well executed and presented study. I have only a few comments.

1) P. 4: "deuterostomes" should be "deuterosomes"

2) Statistics info in legends: the authors should clearly state the number n that was used for statistics. In several cases two numbers (n and N) are given. I suppose one is sample size and one is population size?

3) I suggest to mention in the Results section that centrin-GFP mice were used. It is not obvious from the figures and legends that this was the case.

4) Figure 2: in the text it is stated that most ependymal Plk4 KO cells had 2 centrioles, but quantification in 2A is more consistent with ~10 centrioles; also, I suggest that the authors state that they used acetylated tubulin staining as readout for multi-ciliated cell counting?

5) Related to point 4): in the text and legends it is stated many times that motile cilia were formed or scored. This could be misinterpreted. Since motility was not assessed, I would refer to these simply as multiple cilia, multi-ciliated cells, or similar.

*Reviewer #2 (Recommendations for the authors):*

In my opinion, the data in this manuscript are of outstanding quality, successfully demonstrating the role of PLK4 in centriole amplification and apical expansion in MCCs. Hence, I do not feel the authors need to do any additional experiments to prove their points.

The manuscript is also excellently written. I only have a few points for the authors to consider before proceeding:

– Figure S4A and S4C: it would be good to add statistical analysis to these data.

– Figure S2D and S7A-B: I would entertain the possibility of transferring them to main figures.

– Figure 2E: specify differentiation day in legend.

– Figure S7: title is wrong (it's same as Figure 5, not corresponding to data in Figure S7).

– Figure S4D and S5C: some text has been deleted in these legends.

– Figure 1C and 1E: I would indicate that DEUP1 is a deuterosome marker when it is first used in Figure 1C, rather than later in Figure 1E.

– Figure 1A: covereslips > coverslips

– Page 4, paragraph-2, line-10: deuterostomes > deuterosomes

– Page 9, line-5 from bottom: microtube > microtubule

– Page 10, line-6: define PCM acronym.

---

## [Author Response]

Reviewer #1 (Recommendations for the authors):This is a very well executed and presented study. I have only a few comments.1) P. 4: "deuterostomes" should be "deuterosomes"

This error has been corrected.

2) Statistics info in legends: the authors should clearly state the number n that was used for statistics. In several cases two numbers (n and N) are given. I suppose one is sample size and one is population size?

We added a “statistical analysis” section to the methods to explain that statistics were performed by comparing the average of the biological replicates (N).

3) I suggest to mention in the Results section that centrin-GFP mice were used. It is not obvious from the figures and legends that this was the case.

The following text has been added on page 8; “Some mice also contained a Centrin-GFP transgene to mark centrioles (indicated in figures as Centrin-GFP).”

4) Figure 2: in the text it is stated that most ependymal Plk4 KO cells had 2 centrioles, but quantification in 2A is more consistent with ~10 centrioles;

Figure 2A has been updated to a superplot so that the centriole number in each cell can be easily visualized. >50% of cells have 2 centrioles, and a small fraction of cells had between 5-10 centrioles. These cells may express low levels of PLK4 due to incomplete protein turnover after Cre recombination. The ~20% of cells that had amplified (>11) centrioles are likely to have escaped Cre recombination and express normal levels of PLK4. A breakdown of the percentage of cells with specific centriole numbers is within Author response image 1.

**Author response image 1. sa2fig1:** 

also, I suggest that the authors state that they used acetylated tubulin staining as readout for multi-ciliated cell counting?

The following text has been added to the figure legend for 2C: Acetylated tubulin staining was used to identify multiciliated cells.

5) Related to point 4): in the text and legends it is stated many times that motile cilia were formed or scored. This could be misinterpreted. Since motility was not assessed, I would refer to these simply as multiple cilia, multi-ciliated cells, or similar.

This has been corrected in the text.

Reviewer #2 (Recommendations for the authors):In my opinion, the data in this manuscript are of outstanding quality, successfully demonstrating the role of PLK4 in centriole amplification and apical expansion in MCCs. Hence, I do not feel the authors need to do any additional experiments to prove their points.The manuscript is also excellently written. I only have a few points for the authors to consider before proceeding:– Figure S4A and S4C: it would be good to add statistical analysis to these data.

We have added statistical analysis to Figure S4C to show that the percent recovery after 10 minutes is not statistically different between centrioles in control cells and centrin assemblies in *Plk4^F/F^;Cre^ERT2^* cells. We decided not to add statistical analysis in Figure S4A because we do not have data for the percent of Stage I cells with centrin aggregates, which would be the appropriate comparison for the statistical analysis.

– Figure S2D and S7A-B: I would entertain the possibility of transferring them to main figures.

Following the reviewer’s suggestion, we created a new Figure 3 containing the data previously in Figure S2D. We now include the data from Figure S7A-B in Figure 5.

– Figure 2E: specify differentiation day in legend.

The figure legend for 2E has been updated to specify that these images were taken at differentiation day 5.

– Figure S7: title is wrong (it's same as Figure 5, not corresponding to data in Figure S7).

This has been corrected. The title of Figure S7 is: “Plk4 kinase activity is required for centriole amplification in mTECs.”

– Figure S4D and S5C: some text has been deleted in these legends.

This has been corrected.

– Figure 1C and 1E: I would indicate that DEUP1 is a deuterosome marker when it is first used in Figure 1C, rather than later in Figure 1E.

Figure 1C now clarifies that DEUP1 is a marker for deuterosomes.

– Figure 1A: covereslips > coverslips

This has been corrected, and the figure has been updated.

– Page 4, paragraph-2, line-10: deuterostomes > deuterosomes

This has been corrected.

– Page 9, line-5 from bottom: microtube > microtubule

This has been corrected.

– Page 10, line-6: define PCM acronym.

PCM has been defined; The sentence now reads: “We examined the localization of several other centriole and pericentriolar material (PCM) proteins in mTECs lacking PLK4.”